# A Combinatorial Algorithm for the Semi-Discrete Optimal Transport Problem

**Pankaj K. Agarwal**[1], **Sharath Raghvendra**[2], **Pouyan Shirzadian**[3], and **Keegan Yao**[1]

[1]Duke University, [2]North Carolina State University, [3]Virginia Tech

## Abstract

Optimal Transport (OT, also known as the Wasserstein distance) is a popular metric for comparing probability distributions and has been successfully used in many machine-learning applications. In the semi-discrete 2-Wasserstein problem, we wish to compute the cheapest way to transport all the mass from a continuous distribution $\mu$ to a discrete distribution $\nu$ in $\mathbb{R}^d$ for $d \geq 1$, where the cost of transporting unit mass between points $a$ and $b$ is $d(a, b) = \|a - b\|^2$. When both distributions are discrete, a simple combinatorial framework has been used to find the exact solution (see e.g. [Orlin, STOC 1988]). In this paper, we propose a combinatorial framework for the semi-discrete OT, which can be viewed as an extension of the combinatorial framework for the discrete OT but requires several new ideas. We present a new algorithm that given $\mu$ and $\nu$ in $\mathbb{R}^2$ and a parameter $\varepsilon > 0$, computes an $\varepsilon$-additive approximate semi-discrete transport plan in $O(n^4 \log n \log \frac{1}{\varepsilon})$ time (in the worst case), where $n$ is the support-size of the discrete distribution $\nu$ and we assume that the mass of $\mu$ inside a triangle can be computed in $O(1)$ time. Our algorithm is significantly faster than the known algorithms, and unlike many numerical algorithms, it does not make any assumptions on the smoothness of $\mu$. As an application of our algorithm, we describe a data structure to store a large discrete distribution $\mu$ (with support size $N$) using $O(N)$ space so that, given a query discrete distribution $\nu$ (with support size $k$), an $\varepsilon$-additive approximate transport plan can be computed in $O(k^3 \sqrt{N} \log \frac{1}{\varepsilon})$ time in 2 dimensions. Our algorithm and data structure extend to higher dimensions as well as to $p$-Wasserstein problem for any $p \geq 1$.

## 1 Introduction

Optimal Transport (OT) is a powerful metric for comparing probability distributions and is used in many machine-learning applications. The semi-discrete optimal transport problem asks for the cheapest transport plan to transport mass from (a possibly continuous) distribution $\mu$ that is stored compactly (say, using a deep neural network) to a discrete distribution $\nu$. In recent years, the semi-discrete OT has been used in data mining [31], image processing [28, 29, 33, 40], computational biology [53], variational inference [7], blue noise generation [21, 51], optics [46], solving PDEs [30, 34], and generative models [8, 9, 19].

More formally, a *semi-discrete transport plan* $\tau$ between a continuous probability distribution $\mu$ defined over a compact support $A \subset \mathbb{R}^d$ and a discrete distribution $\nu$ with a support set $B$ of $n$ points in $\mathbb{R}^d$ is a distribution over $A \times B$ whose marginals are dominated by $\mu$ and $\nu$, i.e., $\tau \colon A \times B \to \mathbb{R}_{\geq 0}$ is a *transport plan* between $\mu$ and $\nu$ where $\sum_{b \in B} \tau(a, b) \leq \mu(a)$ for all $a \subseteq A$ and $\int_A \tau(a, b) \, da \leq \nu(b)$ for all $b \in B$. A transport plan $\tau$ is *complete* if $\tau$ transports all mass of $\mu$ to $\nu$. For any fixed $p \geq 1$, in

---

*The authors are listed in alphabetical order.

38th Conference on Neural Information Processing Systems (NeurIPS 2024).

the *semi-discrete p-Wasserstein problem*, the goal is to compute a complete transport plan $\tau$ between $\mu$ and $\nu$ minimizing the cost $\rlap{/}{c}(\tau) := \int_A \sum_{b \in B} \|a - b\|^p \tau(a, b) \, da$. An optimal plan is referred to as a $W_p$-*OT plan*[1]. For any parameter $\varepsilon > 0$, a transport plan $\tau$ is called an $\varepsilon$-*close $W_p$-OT plan* if $\rlap{/}{c}(\tau) \leq \rlap{/}{c}(\tau^*) + \varepsilon$, where $\tau^*$ is a $W_p$-OT plan.

An optimal solution for the semi-discrete OT problem can be compactly represented as the weighted Voronoi diagram (also called the Laguerre diagram) with respect to a weight assignment $y(b)$ on every point $b \in B$ [12]. The choice of weights guarantees that mass at $b$ is equal to the mass of $\mu$ inside the Voronoi cell (also called the Laguerre cell) of $b$ and the optimal transport plan simply transports the mass at $b$ to the mass inside its Voronoi cell. For 2 dimensions, the weighted Voronoi diagram under the squared Euclidean distance (also known as the power diagram) can be constructed in $O(n \log n)$ time [24, 57]. For higher dimensions $d > 2$, the construction time would be $O(n^{\lceil (d+1)/2 \rceil})$ [11].

The compact and connected nature of the semi-discrete optimal transport plans makes them attractive for many ML applications; for instance, they can help achieve stability in training GANs and avoid issues such as discontinuities and mode collapse [10, 19, 55], they improve the mapping between continuous latent spaces and discrete data in Variational Autoencoders (VAEs) [9], and also have been used in diffusion-based generative models [42]. However, there are no known exact algorithms for computing semi-discrete optimal transport. Additionally, computing an $\varepsilon$-close transport plan is known to be $\#P$-Hard with respect to $d$ and $\log 1/\varepsilon$, i.e., an algorithm with an execution time that is polynomial in $d$ and $\log 1/\varepsilon$ seems unlikely [58]. Due to the intractability of the semi-discrete optimal transport in high dimensions, researchers have considered taking $n$ samples from the model, which is a continuous distribution $\mu$, and computing an optimal discrete transport plan between the empirical distribution $\mu_n$ defined on the $n$ samples and $\nu$ [43]. It has been shown that this empirical $p$-Wasserstein distance converges to the true semi-discrete $p$-Wasserstein distance at a rate of $n^{-1/2p}$ [22, 35, 50]; note that the rate of convergence does not depend on the dimension. However, the optimal transport plan from samples is not necessarily a good approximation of the semi-discrete transport plan and may cause biased gradients [14, 19].

For the discrete OT problem, there are several near-optimal scalable exact and approximation algorithms [1, 4, 5, 20, 25, 26, 36, 39, 52, 54], some of which extend to very high dimensions. For the semi-discrete OT in low dimensions, despite extensive work, scalable algorithms to find optimal semi-discrete transport plans remain elusive. There are algorithms to compute an $\varepsilon$-close semi-discrete transport plan using numerical solvers [12, 15, 17, 21, 37, 38, 41, 48], entropic regularization [6, 16, 32], and multiscale approaches [40, 44]. The execution time of all these algorithms is exponential in both $d$ and $\log 1/\varepsilon$. Furthermore, their convergence relies on a smoothness parameter of $\mu$. For instance, a notable numerical algorithm by Oliker and Prussner [48] assumes that for a point $b \in B$, a small change in $y(b)$ will change the mass of $\mu$ inside the Voronoi cell of $b$ by a proportionately small amount [45, Remark 22]. Under this assumption, their algorithm executes $\text{poly}(n, 1/\varepsilon)$ iterations, where each iteration requires the computation of a weighted Voronoi diagram which takes $n^{O(d)}$ time. Their algorithm slows down when $\mu$ is non-smooth and does not even converge when $\mu$ is a discrete distribution. Furthermore, these methods approximate the transport cost but the transport plan that they compute may not be an approximation of the optimal weighted Voronoi diagram.

Recently, Agarwal *et al.* [5] described a *cost-scaling* paradigm to compute an $\varepsilon$-close semi-discrete transport plan. Their algorithm executes $\log \frac{\Delta}{\varepsilon}$ scales, where $\Delta$ is the diameter of $A \cup B$. Within each scale, they create an instance of the discrete OT problem of size $O(n^5)$ in $\mathbb{R}^2$ (and $n^{O(d)}$ in $\mathbb{R}^d$). Using any existing strongly polynomial primal-dual discrete OT solver, such as the algorithm by Orlin [49], their algorithm computes an exact discrete OT plan for each instance and updates the weights for $B$. The OT plan computed in the final scale is an $\varepsilon$-close transport plan. They show that the weight assigned to any point $b \in B$ is $\varepsilon$ away from the optimal weight assignment. The overall execution time of this algorithm is $O(n^9 \log \frac{\Delta}{\varepsilon})$ in $\mathbb{R}^2$ and $n^{O(d)} \log \frac{\Delta}{\varepsilon}$ in $d$-dimensions. Furthermore, in the limit, the transport plan of their algorithm converges to the optimal weighted Voronoi diagram. Note that their algorithm does not make any assumptions on $\mu$.

**Our Contributions.** The following theorem states our main result.

**Theorem 1.1.** *Let $\mu$ be a continuous probability distribution defined on a compact set $A \subset \mathbb{R}^2$, $\nu$ a discrete probability distribution with a support $B \subset \mathbb{R}^2$ of size $n$, and $\varepsilon > 0$ a parameter. Suppose*

---

[1]The $p$-Wasserstein cost of $\tau$ is defined as $\rlap{/}{c}(\tau)^{1/p}$, and the $p$-Wasserstein distance between $\mu$ and $\nu$ is the $p$-Wasserstein cost of a $W_p$-OT plan.

*there exists an oracle that, given a triangle $\varphi$, returns the mass of $\mu$ inside $\varphi$ in $\Phi$ time. Then, an $\varepsilon$-close $W_2$-OT plan between $\mu$ and $\nu$ can be computed in $O(n^3(\Phi + n \log n) \log \frac{\Delta}{\varepsilon})$ time in the worst-case, where $\Delta$ is the diameter of $A \cup B$.*

Similar to the algorithm by Agarwal *et al.* [5], our algorithm is also based on a cost-scaling approach and executes $O(\log \frac{\Delta}{\varepsilon})$ scales. However, the algorithm within each scale is different. Agarwal *et al.* create a discrete OT instance with $n^4$ vertices and $n^5$ edges and use an exact discrete OT algorithm to solve this instance. Instead, we extend the combinatorial primal-dual framework of discrete OT to the continuous space and present an algorithm to find the desired semi-discrete transport plan in $O(n^4 \log n)$ time in the worst case. There are several challenges in extending the combinatorial framework to semi-discrete settings, and overcoming these challenges is one of the main technical contributions of the paper.

In more detail, in each scale, our algorithm maintains a (partial) transport plan, iteratively computes a set of augmenting paths, and augments the transport plan along such paths until all of the mass is transported. In order to assist in finding augmenting paths, we define a residual graph of size $O(n^3)$ in the continuous space. Algorithms for discrete OT maintain dual weights for all points in $A \cup B$. Unlike in the discrete setting where the vertex set is fixed, the vertex set of our residual graph includes "continuous regions", which evolve over time, and the vertex set of the residual graph changes. Therefore, we are able to maintain weights only for points in $B$ and not for the regions in $A$. Our primal-dual framework (especially the definition of admissibility in Section 2), as well as our algorithm (in particular Sections 3.2 and 3.4) contain a number of novel ideas, carefully designed to address various challenges that arise due to the dynamically changing continuous regions of the residual graph. Like all existing algorithms, our algorithm also requires access to an oracle that, given a query triangle, returns the mass of $\mu$ inside the triangle. We note that Dijkstra's shortest path algorithm has been extended to continuous space [47], but we are not aware of any previous work that extends a combinatorial discrete OT framework to continuous space.

Our algorithm extends to any dimension $d \geq 2$ and any $p \geq 1$ in a straightforward way. For $d > 2$, the algorithm in Theorem 1.1 can be shown to have an execution time of $O(n^{d+1}(\Phi + n \log n) \log \frac{\Delta}{\varepsilon})$. Note that the runtime of Oliker-Prussner's algorithm has a factor $1/\varepsilon$ while ours has only $\log 1/\varepsilon$. Unlike their algorithm, ours does not make any assumptions on the smoothness of $\mu$. Furthermore, similar to Agarwal et al. [5], our transport plan approximates an optimal weighted Voronoi diagram within a small additive factor.

One consequence of our algorithm is that we can use it to design a data structure that answers $\varepsilon$-close optimal transport queries efficiently. More precisely, consider a large distribution $\mu$ with support of size $N$. We can preprocess this distribution into a data structure that can return the total weight of points inside a query triangle in $O(N^{1-1/d})$ time [59]. By using Theorem 1.1, we can report an $\varepsilon$-additive approximate transport plan to any query distribution $\nu$ with support of size $k$ in $O(k^{d+1}N^{1-1/d} \log \frac{\Delta}{\varepsilon})$ time. Note that this query time is sub-linear in $N$. There has been some work on computing 1-Wasserstein distance approximately in sub-linear time and for answering nearest neighbor queries under 1-Wasserstein distance [13], but these algorithms have larger error and do not extend to 2-Wasserstein distance.

**Theorem 1.2.** *Let $\mu$ be a discrete probability distribution with a support $A \subset \mathbb{R}^d$ of size $N$. The distribution $\mu$ can be preprocessed, in $O(N \log N)$ time, into an $O(N)$ size data structure so that for a discrete probability distribution $\nu$ with a support $B \subset \mathbb{R}^d$ of size $k$, and a parameter $\varepsilon > 0$, an $\varepsilon$-close $W_2$-OT plan between $\mu$ and $\nu$ can be computed in $O(k^{d+1}(N^{1-1/d} + k \log k) \log \frac{\Delta}{\varepsilon})$ time, where $\Delta$ is the diameter of $A \cup B$.*

If the points in the supports of $\mu$ and $\nu$ have integer coordinates and the masses on them are rational numbers, we can adapt our data structure in Theorem 1.2 to compute an exact $W_p$-OT plan between $\mu$ and $\nu$ in $O(k^{d+1}(N^{1-1/d} + k \log k) \log \frac{\Delta}{\varepsilon})$ time for any fixed even value of $p$.

We also note that in 2-dimensions, for $k < N^{1/4}$, we can use the data structure to compute an exact discrete OT plan for any distribution $\nu$ in $O(k^3 N^{1/2} \log \Delta)$ time that is faster than any existing OT algorithm that takes $(Nk)^{1+o(1)}$ time [3, 18].

For simplicity in presentation, we restrict our presentation to $d = 2$ and $p = 2$.

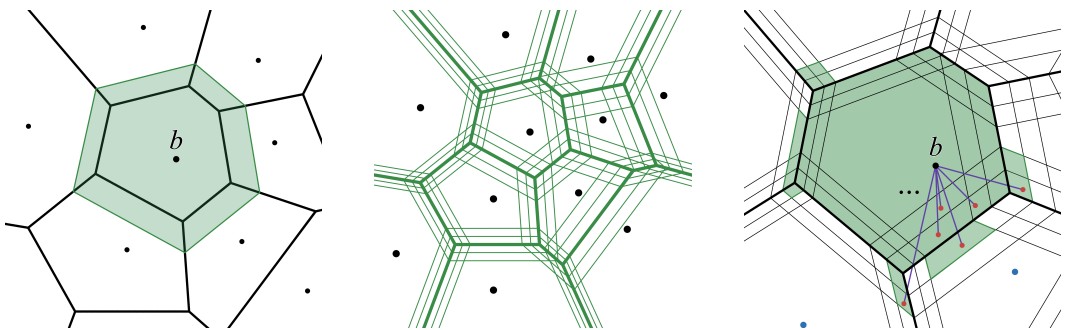

Figure 1: (left) The $\delta$-expanded Voronoi cell $V_b^\delta$ of $b$ (green shaded area), (middle) the partitioning $\mathcal{X}_\delta$, and (right) the green region shows the mass of $\mu$ that is transported to $b$, the red points show the representative points of regions, and the purple segments show the compressed transport plan $\hat{\tau}$.

## 2   Combinatorial Framework

In this section, we extend the combinatorial framework used by discrete OT algorithms to the semi-discrete settings. Let $\mu$ be a continuous probability distribution defined over a compact support $A \subset \mathbb{R}^2$ and $\nu$ be a discrete distribution with a support set $B$ of $n$ points in $\mathbb{R}^2$.

**Weighted Voronoi diagram.** For any pair of points $a, b \in \mathbb{R}^2$, let $\mathrm{d}(a,b) = \|a-b\|^2$. Given a weight function $y : B \to \mathbb{R}_{\geq 0}$ on the points in $B$, the *weighted distance* $\mathrm{d}_y(a,b)$ from a point $b \in B$ to any point $a \in \mathbb{R}^2$ is defined as $\mathrm{d}_y(a,b) = \mathrm{d}(a,b) - y(b)$. For a point $b \in B$, the *Voronoi cell* of $b$ is the locus of all points with $b$ as their weighted nearest neighbor; more formally,

$$\mathrm{Vor}_y(b) = \{a \in \mathbb{R}^2 \mid \mathrm{d}_y(a,b) \leq \mathrm{d}_y(a,b'), \forall b' \in B\}.$$

The *weighted Voronoi diagram* $\mathrm{VD}_y(B)$ of the points $B$ with weights $y(\cdot)$ is the decomposition of $\mathbb{R}^2$ induced by Voronoi cells. There exists a weight function $y(\cdot)$ for $B$ such that $\mu(\mathrm{Vor}_y(b)) = \nu(b)$ for every point $b \in B$, and an optimal semi-discrete transport plan transports the mass of $\nu$ at each point $b \in B$ to the mass of $\mu$ in $\mathrm{Vor}_y(b) \cap A$ [12].

**$\delta$-expanded Voronoi cell.**   Consider a weight function $y(\cdot)$ for the points in $B$. For any point $b \in B$ and a parameter $\delta > 0$, consider the following weight function $y_b^\delta$.

$$y_b^\delta(b') = \begin{cases} y(b') + \delta, & b' = b, \\ y(b'), & b' \neq b. \end{cases} \tag{1}$$

The *$\delta$-expanded Voronoi cell* of $b$, denoted by $V_b^\delta$, is simply the Voronoi cell of $b$ in the weighted Voronoi diagram $\mathrm{VD}_{y_b^\delta}(B)$ of the point set $B$ with weights $y_b^\delta(\cdot)$. See Figure 1 (left).

**$\delta$-feasibility.** Any (possibly partial) transport plan $\tau$ between $\mu$ and $\nu$ along with a weight function $y(\cdot)$ for the points in $B$ is *$\delta$-feasible* if

(F1)  for any pair $(a,b) \in A \times B$ with $\tau(a,b) > 0$, the point $a$ lies inside the $2\delta$-expanded Voronoi cell $V_b^{2\delta}$.

For any $\delta$-feasible transport plan $\tau, y(\cdot)$, if $\tau$ is a complete transport plan between $\mu$ and $\nu$, then $\tau, y(\cdot)$ is called a *$\delta$-optimal transport plan*. Recall that any optimal transport plan transports the mass at $b$ to its weighted Voronoi cell. In a $\delta$-optimal transport plan, however, the mass of each point $b \in B$ is transported inside a $2\delta$-expansion of the Voronoi cell of $b$. This introduces an additive increase of at most $2\delta$ in the cost of the transport plan.

**Lemma 2.1.** *Any $\delta$-optimal transport plan $\tau, y(\cdot)$ between $\mu$ and $\nu$ is $2\delta$-close.*

See Appendix A for a proof. Next, given a $\delta$-feasible transport plan $\tau, y(\cdot)$, we define a residual graph and an augmenting path. We also introduce the process of augmenting $\tau$ along an augmenting path, which allows us to increase the mass transported by $\tau$.

**Residual graph.** Given a $\delta$-feasible (possibly partial) transport plan $\tau, y(\cdot)$, we construct a residual graph $\mathcal{G}_\delta$ by first partitioning the support $A$ of $\mu$ into regions to form the vertex set of $\mathcal{G}_\delta := \mathcal{G}(\tau, y, \delta)$ and then defining a set of directed edges.

*Vertex set.* For each point $b \in B$, consider the Voronoi cell of $b$ and its $\delta$- and $2\delta$-expansions $V_b^\delta$ and $V_b^{2\delta}$. Let $\mathcal{X}_\delta$ denote the arrangement [2] of these $3n$ cells across all $n$ points of $B$. See Figure 1 (middle). For each region $\varphi$ in this arrangement, pick an arbitrary *representative point* $r_\varphi$ inside $\varphi$ and assign it a mass of $\mu_{r_\varphi} := \mu(\varphi)$, where $\mu(\varphi) = \int_\varphi \mu(a)\, da$ denotes the mass of $\mu$ inside $\varphi$. Let $A_\delta$ denote the set of representative points of all regions in $\mathcal{X}_\delta$. The vertex set of $\mathcal{G}_\delta$ is the point set $A_\delta \cup B$ along with a source vertex $s$ and a sink vertex $t$. We refer to any point $b \in B$ whose mass is not fully transported by $\tau$ as a *free* point and define its *excess* as the amount of mass of $b$ that is not transported by $\tau$, i.e., $\mathrm{ex}(b) = \nu(b) - \int_{a \in A} \tau(a, b)\, da$. Similarly, any point $r_\varphi \in A_\delta$ is free if $\tau$ does not fully transport the mass into the region $\varphi$, and its excess is defined as $\mathrm{ex}(r_\varphi) = \mu_{r_\varphi} - \sum_{b' \in B} \tau(\varphi, b')$.

*Edge set.* For each pair $(r_\varphi, b) \in A_\delta \times B$, if $\tau(\varphi, b) > 0$, we add a *backward edge* directed from $r_\varphi$ to $b$ in the residual graph. Furthermore, if $r_\varphi \in V_b^{2\delta}$, we add a *forward edge* directed from $b$ to $r_\varphi$ in $\mathcal{G}_\delta$. Additionally, we add a forward edge from the source $s$ to every free point $b \in B$ and a backward edge directed from every free vertex $r_\varphi$ to $t$. This completes the description of the residual graph.

**Lemma 2.2.** *For any $\delta > 0$ and a $\delta$-feasible transport plan $\hat{\tau}, y(\cdot)$, the residual graph $\mathcal{G}_\delta$ has $O(n^2)$ vertices and $O(n^3)$ edges.*

While describing our algorithm, it is useful to have the definition of weighted distance for all the backward edges, including those incident on $t$. Therefore, we extend the definition of weighted distance to any edge $(r_\varphi, t)$ as follows. Let $b_\varphi$ denote the weighted nearest neighbor of $r_\varphi$ in $B$, i.e., $b_\varphi := \min_{b \in B} d_y(r_\varphi, b)$. Define $d_y(r_\varphi, t) := d_y(r_\varphi, b_\varphi) + \delta$.

**Compressing a semi-discrete transport plan.** Given a semi-discrete transport plan $\tau$ that transports mass from $B$ to $A$, we construct a transport plan $\hat{\tau}$ from $B$ to $A_\delta$ as follows. For each pair $(r_\varphi, b) \in A_\delta \times B$, let $\hat{\tau}(r_\varphi, b) := \tau(\varphi, b)$, i.e., we assign the entire mass transported from $b$ to $\varphi$ to the pair $(r_\varphi, b)$ (Figure 1 (right)). We refer to the transport plan $\hat{\tau}$ as the *compressed transport plan*.

**Lemma 2.3.** *For any $\delta$-feasible semi-discrete transport plan $\tau, y(\cdot)$, the compressed transport plan $\hat{\tau}$ along with weights $y(\cdot)$ is also $\delta$-feasible.*

Conversely, consider a transport plan $\hat{\tau}, y(\cdot)$ from $B$ to $A_\delta$. One can compute a semi-discrete transport plan $\tau$ that, given a pair $(r_\varphi, b) \in A_\delta \times B$, arbitrarily transports a mass of $\hat{\tau}(r_\varphi, b)$ from $b$ to $\varphi$.

**Lemma 2.4.** *Any $\delta$-feasible transport plan $\hat{\tau}, y(\cdot)$ from $B$ to $A_\delta$ can be converted into a $\delta$-feasible semi-discrete transport plan $\tau$ from $B$ to $A$.*

We say that any compressed transport plan $\hat{\tau}$ is a *forest* if the edges transporting a positive mass in $\hat{\tau}$ do not create an undirected cycle.

**Augmentation.** Given the residual graph $\mathcal{G}_\delta$ for a $\delta$-feasible transport plan $\hat{\tau}, y(\cdot)$, an *alternating path* (resp. *alternating cycle*) is a directed path (resp. directed cycle) in $\mathcal{G}_\delta$. Note that, in any directed path (resp. cycle) in the residual graph, the edges alternate between forward and backward edges. An *augmenting path* $P = \langle s, b_1, r_1, \ldots, r_k, t = b_{k+1} \rangle$ is a directed path from the source vertex $s$ to the sink vertex $t$ in the residual graph. By construction, the vertex $b_1$ and the vertex $r_k$ are free vertices in the residual graph. One can *augment* $\hat{\tau}$ along $P$ as follows. Define the bottleneck capacity of the augmenting path $P$ as $\mathrm{bc}(P) := \min\{\mathrm{ex}(b_1), \mathrm{ex}(r_k), \min_{i \in [1, k-1]} \hat{\tau}(r_i, b_{i+1})\}$. To augment $\hat{\tau}$ along $P$, set $\hat{\tau}(r_i, b_i) \leftarrow \hat{\tau}(r_i, b_i) + \mathrm{bc}(P)$ for each forward edge $(b_i, r_i) \in P$ and $\hat{\tau}(r_i, b_{i+1}) \leftarrow \hat{\tau}(r_i, b_{i+1}) - \mathrm{bc}(P)$ for each backward edge $(r_i, b_{i+1}) \in P$.

**Lemma 2.5.** *The transport plan obtained after augmenting a $\delta$-feasible transport plan $\hat{\tau}, y(\cdot)$ along any augmenting path $P$ in the residual graph is $\delta$-feasible.*

Consider the following straightforward way to augment a semi-discrete transport plan. Given a $\delta$-feasible semi-discrete transport plan $\tau, y(\cdot)$, we can compute an augmenting path $P$ in the residual graph and augment the compressed transport plan $\hat{\tau}$ along $P$. From Lemmas 2.3 and 2.5, $\hat{\tau}$ remains $\delta$-feasible and from Lemma 2.4, the updated $\delta$-feasible transport plan $\hat{\tau}$ can be converted to a $\delta$-feasible semi-discrete transport plan as desired. To obtain a complete transport plan, one can iteratively apply this procedure until there are no augmenting paths in the residual graph; however, this may result in an unbounded number of iterations. To obtain an efficient algorithm, we iteratively compute a set of

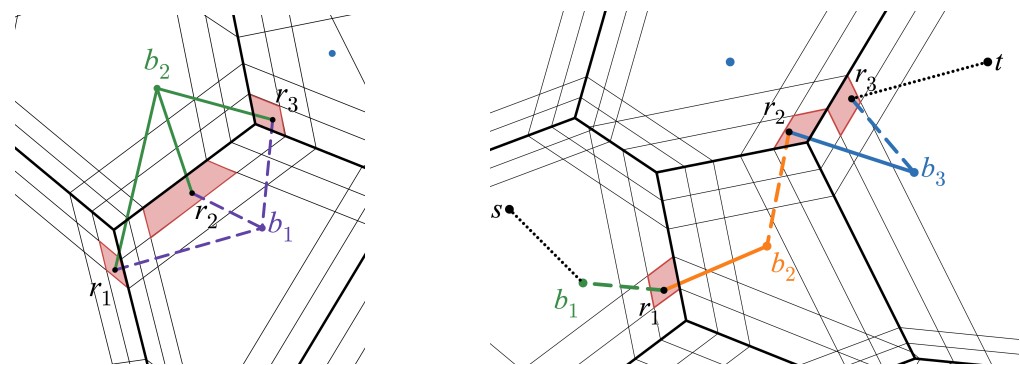

Figure 2: (left) Three admissible triples $(b_1, r_1, b_2)$, $(b_1, r_2, b_2)$, and $(b_1, r_3, b_2)$, where solid (resp. dashed) lines show backward (resp. forward) edges, and (right) an admissible augmenting path.

special augmenting paths called the "admissible" augmenting paths and augment the transport plan along these paths. By doing so, we can reduce the number of iterations to $O(n)$.

**Admissibility.** Suppose $\hat{\tau}, y(\cdot)$ is a $\delta$-feasible transport plan between $\mu$ and $\nu$. For any pair of points $b_1, b_2 \in B$ and any region $\varphi \in \mathcal{X}_\delta$ such that $\hat{\tau}(r_\varphi, b_2) > 0$, the triple $(b_1, r_\varphi, b_2)$ is *admissible* if $d_y(r_\varphi, b_1) < d_y(r_\varphi, b_2)^2$. See Figure 2 (left). Note that an admissible triple $(b_1, r_\varphi, b_2)$ forms by a forward edge followed by a backward edge in the residual graph satisfying $d_y(r_\varphi, b_1) < d_y(r_\varphi, b_2)$. Intuitively, for any admissible triple $\langle b_1, r_\varphi, b_2 \rangle$, the mass of $\mu$ inside $r_\varphi$ is transported from $b_2$ but $b_1$ is nearer to $r_\varphi$ than $b_2$ (with respect to the weighted distances).

We extend the definition of admissibility to augmenting paths and alternating cycles as follows. Any augmenting path (resp. alternating cycle) $P = \langle b_1, r_1, b_2, \ldots, r_k, b_{k+1} \rangle$ is *admissible* if all triples $(b_i, r_i, b_{i+1})$, $i \in [1, k]$, are admissible. In Figure 2(right), $b_2$ transports mass to $r_1$, and $b_1$ is its weighted nearest neighbor. By augmenting along this admissible path, we increase $\hat{\tau}(r_1, b_1)$ and reduce $\hat{\tau}(r_1, b_2)$, thereby transporting more mass to $r_1$ from its weighted nearest neighbor.

## 3 Algorithm

In this section, we present our cost-scaling algorithm that uses the combinatorial framework from Section 2 to compute an $\varepsilon$-close semi-discrete OT plan. Classical discrete OT algorithms assign weights to points in $A \cup B$ and use them to identify a large set of augmenting paths. Their efficiency critically relies on the acyclicity of the "search" graph. In contrast, during the execution of our algorithm, a change in the weights of $B$ creates a new weighted Voronoi diagram, which in turn changes $A_\delta$, the discrete representation of $A$, and thus the vertex set of the residual graph. Since $A_\delta$ may change significantly in each iteration during the execution of our algorithm, we cannot maintain weights for them. This creates significant challenges as the algorithm searches for augmenting paths (See Section 3.2 for details). Furthermore, the updated residual graph may have cycles. We introduce additional steps in our algorithm to eliminate these cycles (See Section 3.4 for details). First, we present an overview of our algorithm and then present the details.

### 3.1 Overview

The algorithm runs for $O(\log \frac{\Delta}{\varepsilon})$ scales, where $\Delta$ is the diameter of $A \cup B$. In each scale $\delta$, our algorithm maintains a transport plan $\hat{\tau}_\delta$ and a weight $y(b)$ for every point $b \in B$. Initially, set $\delta = \Delta^2$ and define $y(b) = 0$ for all $b \in B$. In each scale $\delta$, execute the following steps.

1. *Initialization:* Set $\tau_\delta$ to be an empty transport plan. Compute the residual graph $\mathcal{G}_\delta$ and the compressed transport plan $\hat{\tau}_\delta$ with respect to $\tau_\delta, y(\cdot)$.

2. *Iterations:* While $\hat{\tau}_\delta$ is not a complete transport plan:

---

[2]Discrete OT algorithms use the weights assigned to $A \cup B$ to define admissible edges. Since $A_\delta$ evolves during the execution of our algorithm, we cannot maintain weights for them, forcing us to define admissibility on a sequence of two consecutive edges $(b_1, r_\varphi)$ and $(r_\varphi, b_2)$ rather than per edge.

(i) Compute a set of admissible augmenting paths in the residual graph $\mathcal{G}_\delta$ and augment $\hat{\tau}_\delta$ along these paths using the SEARCHANDAUGMENT procedure described in Section 3.2. At the end of this step, there are no admissible augmenting paths in the residual graph.

(ii) Adjust the weights of all points of $B$ that are reachable from the source by admissible paths by $\delta$ and recompute the set $A_\delta$, the residual graph $\mathcal{G}_\delta$, and the compressed transport plan $\hat{\tau}_\delta$ using the INCREASEWEIGHTS procedure described in Section 3.3.

(iii) Update the compressed transport plan $\hat{\tau}_\delta$ and the residual graph using the ACYCLIFY procedure described in Section 3.4, so that the transport plan $\hat{\tau}_\delta$ is a forest and the residual graph does not have any admissible cycles.

3. *Scale Update:* Set $\delta \leftarrow \delta/2$.

After the execution of a scale $\delta \le \varepsilon/2$, our algorithm terminates by returning a complete semi-discrete transport plan $\tau_\delta$ obtained from the compressed complete transport plan $\hat{\tau}_\delta$ (using Lemma 2.4).

**Invariants.** As shown in Section 4 below, our algorithm iteratively updates the weights $y(\cdot)$ and the transport plan $\hat{\tau}_\delta$ while maintaining the following invariants:

(I1) The transport plan $\hat{\tau}_\delta, y(\cdot)$ is $\delta$-feasible, and

(I2) At the start of each iteration, the transport plan $\hat{\tau}_\delta$ is a forest and there are no admissible cycles in the residual graphs.

**Remark.** The algorithm by Agarwal et al. [5] creates a discrete instance in each scale of the algorithm by computing the arrangement of the $\delta$-, $2\delta$-, ..., and $(4n + 1)\delta$-expanded Voronoi cells of each point $b \in B$. Instead of using such a fine partition to create a discrete instance with $O(n^5)$ edges, we work directly with the continuous space, maintain a much smaller residual graph with $O(n^3)$ edges, and use our semi-discrete combinatorial framework to find a transport plan.

## 3.2 SEARCHANDAUGMENT Procedure

The SEARCHANDAUGMENT procedure executes a partial DFS-style search to identify a set of admissible augmenting paths and augments the transport plan along these paths. The SEARCHAN-DAUGMENT procedure is somewhat similar in style to the blocking flow procedure in Dinic's max-flow algorithm [23] or the partial-DFS procedure in Gabow-Tarjan's algorithm [27], both of which rely on the property that there are no admissible cycles in the residual graph. Unlike these algorithms, in our case, if we augment along an arbitrary admissible augmenting path, we may create an admissible cycle. See Figure 3.

We overcome this challenge by carefully calibrating the search algorithm in two ways. First, we begin our search from the free regions instead of the free points of $B$. Thus, we reverse the direction of all the edges of the residual graph and begin our search from the sink $t$. Second, we explore all forward edges incident on a region in the increasing order of their weighted distance. This order of processing edges ensures that no admissible cycles are created and that there are no more admissible augmenting paths in the residual graph after the SEARCHANDAUGMENT procedure terminates. We provide the details below.

Let $\overleftarrow{\mathcal{G}}_\delta$ be the graph formed by reversing the direction of all the edges of $\mathcal{G}_\delta$. We conduct our search starting from the sink $t$ in the graph $\overleftarrow{\mathcal{G}}_\delta$. Initially, mark all points of $B$ and all backward edges as unvisited, define $U := B$ as the set of unvisited points of $B$, and $Q = \langle t = b_0 \rangle$ as the search path that the procedure grows. Execute the following steps until the search path $Q$ becomes empty.

1. If $Q = \langle t = b_0, r_1, b_1, \ldots, r_i, b_i \rangle$ for some $i \ge 0$,

   (a) If there is an edge from $b_i$ to $s$ in $\overleftarrow{\mathcal{G}}_\delta$ (i.e., $b_i$ is a free point), then $P = \langle s, b_i, r_i, \ldots, r_1, t \rangle$ is an augmenting path in $\mathcal{G}_\delta$. Augment $\hat{\tau}_\delta$ along $P$ and set $Q = \langle t = b_0 \rangle$.

   (b) Assume there is not edge from $b_i$ to $s$. If there is an unvisited edge $(b_i, r)$ in $\overleftarrow{\mathcal{G}}_\delta$, then add $r = r_{i+1}$ to $Q$. Otherwise, mark $b_i$ as visited and remove $b_i$ from $U$ and $Q$.

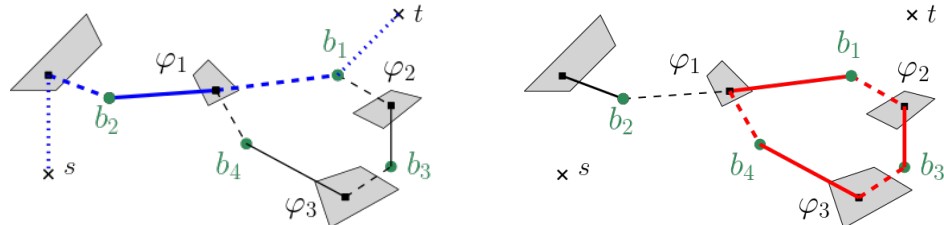

Figure 3: (left) An arbitrary admissible augmenting path $\langle s, b_1, r_{\varphi_1}, b_2 \rangle$ (blue path), and (right) an admissible cycle $\langle b_1, r_{\varphi_2}, b_3, r_{\varphi_3}, b_4, r_{\varphi_1} \rangle$ (red cycle) formed after augmentation.

2. If $Q = \langle t = b_0, r_1, b_1, \ldots, b_i, r_{i+1} \rangle$ for some $i \geq 0$, let $b := \arg\min_{b' \in U} d_y(r_{i+1}, b')$ denote the unvisited point with the minimum weighted distance to $r_{i+1}$[3].

   (a) If $(b, r_{i+1}, b_i)$ is admissible, i.e., $d_y(r_{i+1}, b) < d_y(r_{i+1}, b_i)$, then add $b$ as $b_{i+1}$ to $Q$.
   (b) Otherwise, remove $r_{i+1}$ from $Q$ and mark the edge $(b_i, r_{i+1})$ as visited.

The algorithm terminates when the search path $Q$ becomes empty, i.e., the procedure marked $t$ as visited and removed it from $Q$. The following lemma shows the properties of the SEARCHANDAUGMENT procedure.

**Lemma 3.1.** *Suppose invariants (I1) and (I2) hold at the start of the* SEARCHANDAUGMENT *procedure. Then, during the execution of the* SEARCHANDAUGMENT *procedure,*

   *(S1) the transport plan $\hat{\tau}_\delta, y(\cdot)$ remains $\delta$-feasible,*
   *(S2) any point $b \in B$ (resp. backward edge $(r, b)$) marked as visited will not form an admissible augmenting path during the execution of the procedure, and*
   *(S3) there are no admissible cycles in the residual graph.*

### 3.3 INCREASEWEIGHTS Procedure

The INCREASEWEIGHTS procedure adjusts the weights of $B$ leading to the formation of admissible augmenting paths. The INCREASEWEIGHTS procedure performs a DFS on the residual graph $\mathcal{G}_\delta$ to compute a set $\mathcal{K}$ of all vertices of $B$ that are reachable from the source vertex $s$ by admissible paths. It increases the weights of all points in $\mathcal{K}$ by $\delta$ (expand their Voronoi cells). Recall that the set $A_\delta$, and thus $\mathcal{G}_\delta$ as well as the compressed transport plan $\hat{\tau}_\delta$ depend on the weights of $B$. The procedure then recomputes $A_\delta$, $\mathcal{G}_\delta$, and $\hat{\tau}_\delta$ from $\tau_\delta$. See Appendix B.2 for complete details of the INCREASEWEIGHTS procedure. The following lemma states the properties of the INCREASEWEIGHTS procedure.

**Lemma 3.2.** *Suppose invariant (I1) holds at the start of the* INCREASEWEIGHTS *procedure. Then, during the execution of the* INCREASEWEIGHTS *procedure,*

   *(W1) the transport plan $\hat{\tau}_\delta, y(\cdot)$ remain $\delta$-feasible,*
   *(W2) the weight of each free point $b \in B$ increases by $\delta$, and*
   *(W3) the weight of each point $b \in B$ with free regions inside $V_b^\delta$ remains unchanged.*

### 3.4 ACYCLIFY Procedure

The change in the weights of $B$ in the INCREASEWEIGHTS procedure requires us to recompute the residual graph and the compressed transport plan. This recomputation may potentially create a cycle in the transport plan or an admissible cycle in $\mathcal{G}_\delta$. The ACYCLIFY procedure eliminates such cycles and ensures that the invariant (I2) holds at the start of the next iteration. Converting the transport plan into a forest is critical for the efficiency of the SEARCHANDAUGMENT procedure while eliminating admissible cycles is essential for the correctness of the SEARCHANDAUGMENT procedure. The procedure executes the following steps. First, use the dynamic tree structure by Sleator and Tarjan [56] to make the transport plan $\hat{\tau}_\delta$ a forest. Then, execute a partial DFS search from

---

[3]To perform this step efficiently, in the construction of the residual graph, for each $r \in A_\delta$, we store the list of all neighbors of $r$ sorted in increasing order of their weighted distance.

each unvisited point $b \in B$ similar to the one described in the SEARCHANDAUGMENT procedure to detect admissible cycles. Upon finding an admissible cycle, cancel the cycle right away, remove the vertices of the cycle from the search path, and continue the search. When all vertices are visited, no admissible cycles are remaining in the residual graph. Since canceling admissible cycles could have introduced new cycles in the transport plan, repeat the first step to update $\hat{\tau}_\delta$ and make it a forest. See Appendix B.3 for details. The following lemma shows the properties of the ACYCLIFY procedure.

**Lemma 3.3.** *Suppose invariant (I1) holds at the start of the* ACYCLIFY *procedure. Then, during the execution of the* ACYCLIFY *procedure,*

> *(A1) the transport plan $\hat{\tau}_\delta, y(\cdot)$ remains $\delta$-feasible, and*
> *(A2) the transport plan $\hat{\tau}_\delta$ is a forest and there are no admissible cycles in the residual graph.*

This completes the description of our algorithm.

## 4 Algorithm Analysis

In this section, we first prove the correctness of our algorithm and then analyze its runtime.

**Proof of invariants (I1) and (I2).** For any scale $\delta$, the initial transport plan $\hat{\tau}_\delta$ is empty. Therefore, $\hat{\tau}_\delta$ along with the weights $y(\cdot)$ is $\delta$-feasible. By properties (S1), (W1), and (A1), the transport plan $\hat{\tau}_\delta, y(\cdot)$ remains $\delta$-feasible in each iteration of our algorithm, and therefore, invariant (I1) holds. The invariant (I2) is a direct consequence of property (A2) in Lemma 3.3.

**Proof of Correctness.** From Invariant (I1), in each scale $\delta$, our algorithm maintains a $\delta$-feasible transport plan $\hat{\tau}_\delta, y(\cdot)$ during its execution. The while loop in Step 2 breaks when $\hat{\tau}_\delta$ is a complete transport plan. Therefore, $\hat{\tau}_\delta$ along with weights $y(\cdot)$ is $\delta$-optimal. From Lemma 2.4, one can convert $\hat{\tau}_\delta$ into a $\delta$-optimal semi-discrete transport plan $\tau_\delta$. Given that our algorithm terminates when $\delta \le \varepsilon/2$, from Lemma 2.1, the transport plan returned by our algorithm is $\varepsilon$-close, as desired.

**Efficiency of the algorithm.** The SEARCHANDAUGMENT procedure runs a partial DFS on the residual graph with $O(n^3)$ edges. Upon finding an augmenting path $P$, the procedure augments the transport plan along $P$ in $O(|P|)$ time. In Lemma C.1, we use invariant (I2) to show that the total length of all augmenting paths found by the procedure is $O(n^3)$. Hence, the running time of the SEARCHANDAUGMENT procedure is $O(n^3)$. The INCREASEWEIGHTS procedure stores a sorted list of neighbors for each region and executes a DFS procedure in the residual graph, which takes $O(n^3 \log n)$ time. After increasing the weight of a subset of points of $B$, the procedure recomputes the residual graph $\mathcal{G}_\delta$ and the transport plan $\hat{\tau}_\delta$, which can be done in $O(n^2(\Phi + n \log n))$ time. See Lemma C.2 in the appendix for details. In the ACYCLIFY procedure, converting a transport plan into a forest can be done using a dynamic tree data structure [56] in $O(n^3 \log n)$ time. To eliminate the admissible cycles, the ACYCLIFY procedure relies on a partial DFS that runs in $O(n^3)$ time. Therefore, the total time by the ACYCLIFY procedure is $O(n^3 \log n)$ (see Lemma C.3 for details). Combining the running times of all three procedures, each iteration of step 2 of our algorithm takes $O(n^2(\Phi + n \log n))$ time. In the following lemma, we show that in each scale, the total number of iterations of step 2 of our algorithm is at most $O(n)$.

**Lemma 4.1.** *For each scale $\delta$, the total number of iterations of step 2 of our algorithm is $O(n)$.*

*Proof Sketch.* Let $\tau_{2\delta}, y_{2\delta}(\cdot)$ denote the $2\delta$-feasible semi-discrete transport plan computed by our algorithm for scale $2\delta$, and let $\tau_\delta, y_\delta(\cdot)$ denote a partial semi-discrete transport plan maintained during the execution of step 2 of our algorithm. Let $\mathcal{X}_{2\delta}$ (resp. $\mathcal{X}_\delta$) denote the partitioning of the set $A$ with respect to weights $y_{2\delta}(\cdot)$ (resp. $y_\delta(\cdot)$). Let $\mathcal{Y}$ be the arrangement of all $3n$ cells used to construct $\mathcal{X}_{2\delta}$ with all $3n$ cells used to construct $\mathcal{X}_\delta$. Let $\hat{\tau}_{2\delta}$ (resp. $\hat{\tau}_\delta$) denote the compressed transport plan for $\tau_{2\delta}$ (resp. $\tau_\delta$) using the partitioning $\mathcal{Y}$. It is well-known that one can transform $\hat{\tau}_\delta$ to $\hat{\tau}_{2\delta}$ by augmenting $\hat{\tau}_\delta$ along a set of augmenting paths $\mathcal{P}$ on $\mathcal{Y} \times B$ and rearrange the transported mass along a set of cycles $\mathcal{C}$ on $\mathcal{Y} \times B$. Consider an augmenting path $P = \langle r_1, b_1, \ldots, r_k, b_k \rangle$ in $\mathcal{P}$. Since $P$ is a simple path, it contains each point of $B$ at most once and therefore, it has a length at most $2n - 1$. Additionally, for all $i \in [1, k]$, $\hat{\tau}_{2\delta}(r_i, b_i) > 0$ and for each $i \in [2, k]$, $\hat{\tau}_\delta(r_i, b_{i-1}) > 0$. For each edge $(r_i, b_i)$, since $\hat{\tau}_{2\delta}(r_i, b_i) > 0$, by $2\delta$-feasibility of $\tau_{2\delta}, y_{2\delta}(\cdot)$ (condition (F1)), the point $b_i$ is a $4\delta$-weighted nearest neighbor of $r_i$ with respect to weights $y_{2\delta}(\cdot)$. Similarly, for each edge $(b_{i-1}, r_i)$, since $\hat{\tau}_\delta(r_i, b_{i-1}) > 0$, by $\delta$-feasibility of $\tau_\delta, y_\delta(\cdot)$ (condition (F1)), the point $b_{i-1}$ is a $2\delta$-weighted nearest neighbor of $r_i$ with respect to weights $y_\delta(\cdot)$. Since the length of $P$ is at most $2n - 1$ and the

weight of $b_1$ does not change (by property (W3) in Lemma 3.3), we show that for the free point $b_k$ in $P$, $y_\delta(b_k) - y_{2\delta}(b_k) \leq 6n\delta$. Our algorithm increases the weight of $b_k$ by $\delta$ in each iteration (property (W2) in Lemma 3.3), and therefore after $6n$ iterations, the point $b_k$ cannot be free, i.e., after $O(n)$ iterations, all points of $B$ are fully transported in $\hat{\tau}_\delta$. We provide the full proof in Section C.4.

Using Lemma 4.1, the total time spent in step 2 in each scale of our algorithm is $O(n^3(\Phi + n \log n))$. Since there are $O(\log \frac{\Delta}{\varepsilon})$ scales, the overall runtime is $O(n^3(\Phi + n \log n) \log \frac{\Delta}{\varepsilon})$, thereby proving Theorem 1.1.

## 5 Applications to the Discrete Optimal Transport Problem

In this section, we extend our combinatorial semi-discrete OT algorithm to the discrete OT problem and design a data structure that preprocesses and stores a large discrete distribution $\mu$ and efficiently computes an $\varepsilon$-close OT plan between $\mu$ and any query distribution $\nu$ in sub-linear time relative to the support size of $\mu$. More precisely, given a discrete distribution $\mu$ with a (possibly large) support $A$ of $N$ points in $\mathbb{R}^2$, we design a data structure that, given a query discrete distribution $\nu$ with a support $B$ of $k$ points, computes an $\varepsilon$-close transport plan between $\mu$ and $\nu$ in $O(k^3(\sqrt{N} + k \log k) \log \frac{\Delta}{\varepsilon})$ time. Additionally, we show that if the support points have bounded integer coordinates and the masses are rational numbers, our data structure can efficiently compute an exact discrete OT plan.

At a high level, our data structure interprets the large discrete distribution $\mu$ as a continuous distribution and uses a simplex range-searching data structure as an oracle to compute the mass of $\mu$ inside a query triangle. In this way, for any query distribution $\nu$, one can execute the steps of our combinatorial semi-discrete algorithm to compute an $\varepsilon$-close transport plan between $\mu$ and $\nu$. More formally, our data structure preprocesses the distribution $\mu$ into a simplex range-searching data structure RS-DS, which takes $O(N)$ space, can be built in $O(N \log N)$ time, and returns the mass of $\mu$ inside a query triangle in $\Phi = O(\sqrt{N})$ time [59]. Given a query discrete distribution $\nu$, one can use our algorithm from Section 3 in conjunction with the RS-DS to compute an $\varepsilon$-close transport plan between $\mu$ and $\nu$ in $O(k^3(\sqrt{N} + k \log k) \log \frac{\Delta}{\varepsilon})$ time leading to Theorem 1.2.

Consider the special case where the points in $A \cup B$ have positive integer coordinates bounded by $\lambda$, the mass of $\mu$ (resp. $\nu$) on each point $a \in A$ (resp. $b \in B$) is a rational number of form $\frac{x_a}{T}$ (resp. $\frac{x_b}{T}$) for positive integers $T$ and $x_a$ (resp. $x_b$), and $p$ is an even number. In this case, the $p$-Wasserstein cost of any transport plan between $\mu$ and $\nu$ is an integer multiple of $\frac{1}{T}$, and therefore, any $\frac{1}{2T}$-close transport plan between $\mu$ and $\nu$ would have a minimum cost. Thus, one can compute an exact discrete OT plan between $\mu$ and $\nu$ by setting $\varepsilon = \frac{1}{2T}$ in our data structure, which would have a query time of $O(k^3(\sqrt{N} + k \log k) \log(\lambda T))$, leading to the following corollary.

**Corollary 5.1.** *When the points in the supports of the distributions $\mu$ and $\nu$ have integer coordinates bounded by $\lambda$ and the mass on each point is a rational number of form $\frac{x}{T}$, our data structure computes, for any even number $p$, an optimal solution for the $p$-Wasserstein problem between $\mu$ and $\nu$ in $O(k^3(\sqrt{N} + k \log k) \log(\lambda T))$ time.*

## 6 Conclusion

In this paper, we designed a novel combinatorial framework for the semi-discrete optimal transport problem and used it to compute an $\varepsilon$-close semi-discrete transport plan. We also used this framework to design a data structure that stores a discrete distribution $\mu$ over a large support of size $N$ and can compute $\varepsilon$-close OT cost between $\mu$ and a query discrete distribution $\nu$ in a time that is sub-linear in $N$. We conclude with the following question: Can we use our combinatorial framework to compute an $\varepsilon$-close semi-discrete transport plan between high dimensional distributions in $\text{poly}(n, d, 1/\varepsilon)$ time.

## Acknowledgement

Work by P.A. and K.Y. was supported by NSF grants CCF-22-23870 and IIS-24-02823 and by BSF grant 2022131. Work by S.R. and P.S. was supported by NSF grant CCF-2223871. We thank the anonymous reviewers for their useful comments.

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

# A    Missing Proofs of Section 2

In this section, we present the missing proofs and details of our combinatorial framework described in Section 2.

**Lemma 2.1.** *Any $\delta$-optimal transport plan $\tau, y(\cdot)$ between $\mu$ and $\nu$ is $2\delta$-close.*

*Proof.* Define the weighted cost of a transport plan $\tau'$ as $\cent_y(\tau') := \sum_{b \in B} \int_A \mathrm{d}_y(a,b)\tau'(a,b)\, da$. For any complete transport plan $\tau'$,

$$\cent_y(\tau') = \sum_{b \in B} \int_A \mathrm{d}_y(a,b)\tau'(a,b)\, da = \sum_{b \in B} \int_A \big(\mathrm{d}(a,b) - y(b)\big)\tau'(a,b)\, da = \cent(\tau') - \sum_{b \in B} y(b)\nu(b).$$
$$(2)$$

For any point $a \in A$, let $b_a$ denote the weighted nearest neighbor of $a$, i.e., $b_a := \arg\min_{b \in B} \mathrm{d}_y(a,b)$. By property (F1) in the definition of $\delta$-feasibility,

$$\cent_y(\tau) = \sum_{b \in B} \int_A \mathrm{d}_y(a,b)\tau(a,b)\, da \leq \sum_{b \in B} \int_A \big(\mathrm{d}_y(a,b_a) + 2\delta\big)\tau(a,b)\, da$$

$$= \sum_{b \in B} \int_A \mathrm{d}_y(a,b_a)\tau(a,b)\, da + 2\delta. \tag{3}$$

Let $\tau^*$ denote an optimal transport plan from $\mu$ to $\nu$. Then,

$$\cent_y(\tau^*) = \sum_{b \in B} \int_A \mathrm{d}_y(a,b)\tau^*(a,b)\, da \geq \sum_{b \in B} \int_A \mathrm{d}_y(a,b_a)\tau^*(a,b)\, da. \tag{4}$$

Combining Equations (3) and (4) and plugging $\tau$ and $\tau^*$ in Equation (2),

$$\cent(\tau) = \cent_y(\tau) + \sum_{b \in B} y(b)\nu(b) \leq \left[ \sum_{b \in B} \int_A \mathrm{d}_y(a,b_a)\tau(a,b)\, da + 2\delta \right] + \sum_{b \in B} y(b)\nu(b)$$

$$\leq \cent_y(\tau^*) + 2\delta + \sum_{b \in B} y(b)\nu(b) = \cent(\tau^*) + 2\delta.$$

Therefore, $\cent(\tau) \leq \cent(\tau^*) + 2\delta$ and $\tau$ is $2\delta$-close. $\qquad\square$

**Residual graph.**    Next, we show that for any $\delta > 0$ and any $\delta$-feasible transport plan $\hat{\tau}, y(\cdot)$, the residual graph $\mathcal{G}_\delta$ has $O(n^2)$ vertices and $O(n^3)$ edges, leading to Lemma 2.2. To do so, we show below that the partitioning $\mathcal{X}_\delta$ consists of $O(n^2)$ regions. We then conclude that the number of vertices of $\mathcal{G}_\delta$ is $O(n^2)$. Furthermore, since the residual graph is a bipartite graph between set $B$ of size $n$ and set $A_\delta$ of size $O(n^2)$, the number of edges of $\mathcal{G}_\delta$ would be at most $O(n^3)$.

Recall that the partitioning $\mathcal{X}_\delta$ is constructed as the arrangement of all Voronoi cells, $\delta$-expansions, and $2\delta$-expansions of the Voronoi cells of all points in $B$. Let $\mathcal{V}$ denote the set of vertices of this arrangement. Since the arrangement is a planar graph, the number of faces (i.e., regions) in $\mathcal{X}_\delta$ is $O(|\mathcal{V}|)$. Therefore, to show that $\mathcal{X}_\delta$ has $O(n^2)$ regions, we show that the number of vertices of this arrangement is $O(n^2)$.

For each point $b \in B$, let $V_b^0$ (resp. $V_b^\delta$, $V_b^{2\delta}$) denote the Voronoi cell (resp. $\delta$-expanded Voronoi cell, $2\delta$-expanded Voronoi cell) of $b$, and let $y_b^0$ (resp. $y_b^\delta, y_b^{2\delta}$) denote the weights of $B$ used to compute the cell. Note that each Voronoi cell $V_b^i$ for each $b \in B$ and $i \in \{0, \delta, 2\delta\}$ has at most $n$ vertices. Hence, the total number of Voronoi vertices in the arrangement $\mathcal{X}_\delta$ is $O(n^2)$. Next, we count the number of intersection points of these Voronoi cells. For each pair of points $b_1, b_2 \in B$, consider the pair of cells $V_{b_1}^i$ and $V_{b_2}^j$, for some $i, j \in \{0, \delta, 2\delta\}$. We show that $V_{b_1}^i$ and $V_{b_2}^j$ intersect each other in at most two points.

Define the *weighted bisector* of two points $b$ and $b'$ with respect to weights $y(\cdot)$ as the locus of points that have the same weighted distance to $b$ and $b'$, i.e., all points $x \in \mathbb{R}^2$ such that $\mathrm{d}_y(x,b) = \mathrm{d}_y(x,b')$. Note that under the squared Euclidean distances, the weighted bisector of two points is a straight line. Let $v$ denote an intersection point of $V_{b_1}^i$ and $V_{b_2}^j$, and suppose $v$ lies on the segment of $V_{b_1}^i$

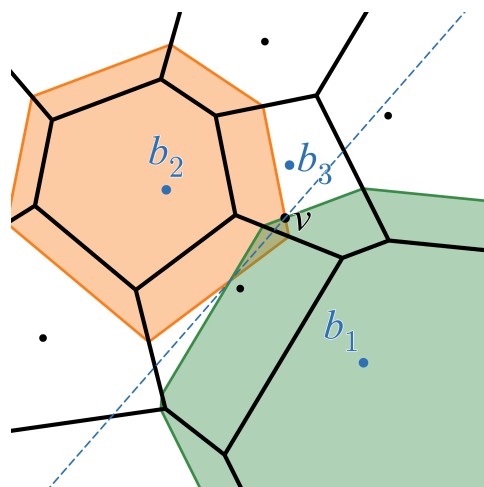

Figure 4: For two cells $V_{b_1}^i$ (green cell) and $V_{b_2}^j$ (orange cell), any intersection point $v$ lies on their weighted bisector.

corresponding to the weighted bisector of $b_1$ and a point $b_3 \in B$ (See Figure 4). Note that if $b_3 = b_2$, then $V_{b_1}^i$ and $V_{b_2}^j$ share a segment containing $v$, which means that

$$\mathrm{d}_y(v, b_1) - i = \mathrm{d}_{y_{b_1}^i}(v, b_1) = \mathrm{d}_{y_{b_1}^i}(v, b_2) = \mathrm{d}_y(v, b_2),$$

and also

$$\mathrm{d}_y(v, b_2) - j = \mathrm{d}_{y_{b_2}^j}(v, b_2) = \mathrm{d}_{y_{b_2}^j}(v, b_1) = \mathrm{d}_y(v, b_1).$$

Therefore, in this case, $i = j = 0$, and the two endpoints of the shared segment is counted toward the number of Voronoi vertices of the arrangement. Hence, we assume $b_3 \neq b_2$. In this case,

$$\mathrm{d}_{y_{b_1}^i}(v, b_1) = \mathrm{d}_{y_{b_1}^i}(v, b_3) = \min_{b \in B} \mathrm{d}_{y_{b_1}^i}(v, b). \tag{5}$$

By the construction of the weight function $y_{b_1}^i(\cdot)$ in Equation (1), for any point $b \in B \setminus \{b_1, b_2\}$, $y_{b_1}^i(b) = y_{b_2}^j(b) = y(b)$; thus, using Equation (5),

$$\mathrm{d}_{y_{b_2}^j}(v, b_3) = \mathrm{d}_{y_{b_1}^i}(v, b_3) = \min_{b \in B \setminus \{b_1, b_2\}} \mathrm{d}_{y_{b_1}^i}(v, b) = \min_{b \in B \setminus \{b_1, b_2\}} \mathrm{d}_{y_{b_2}^j}(v, b). \tag{6}$$

Using Equation (6), the point $v$ also lies in the segment of $V_{b_2}^j$ that corresponds to the weighted bisector of $b_2$ and $b_3$ with respect to weights $y_{b_2}^j$. Thus,

$$\mathrm{d}_{y_{b_1}^i}(v, b_1) = \mathrm{d}_{y_{b_1}^i}(v, b_3) = \mathrm{d}_{y_{b_2}^j}(v, b_3) = \mathrm{d}_{y_{b_2}^j}(v, b_2).$$

In other words, the point $v$ satisfies

$$\mathrm{d}(v, b_1) - (y(b_1) + i) = \mathrm{d}(v, b_2) - (y(b_2) + j),$$

i.e., the point $v$ lies on the weighted bisector of $b_1$ and $b_2$ (the blue dashed line in Figure 4). Since the weighted bisector is a straight line, it intersects the convex polygon $V_{b_1}^i$ in at most 2 points; hence, the two Voronoi cells $V_{b_1}^i$ and $V_{b_2}^j$ intersect each other in at most 2 points. Since the are $O(n^2)$ pairs of such Voronoi cells, each intersecting each other in at most 2 points, the total number of intersection points is at most $O(n^2)$, as claimed.

**Lemma 2.2.** *For any $\delta > 0$ and a $\delta$-feasible transport plan $\hat{\tau}, y(\cdot)$, the residual graph $\mathcal{G}_\delta$ has $O(n^2)$ vertices and $O(n^3)$ edges.*

Next, we extend our bounds for the number of vertices and edges of the residual graph in Lemma 2.2 to any dimension $d \geq 3$ and show that for any pair of $d$-dimensional distributions $\mu$ and $\nu$ and any $\delta$-feasible transport plan $\hat{\tau}, y(\cdot)$, the residual graph $\mathcal{G}_\delta$ has $O(n^d)$ vertices and $O(n^{d+1})$ edges. We prove our bounds by showing that the number of vertices of the arrangement $\mathcal{X}_\delta$ is $O(n^d)$. Using this

bound, we get that the number of regions in the arrangement is upper-bounded by $O(n^d)$; therefore, $|A_\delta| = O(n^d)$ and the residual graph has $O(n^d)$ vertices. Furthermore, since $\mathcal{G}_\delta$ is a bipartite graph between set $B$ with $n$ points and set $A_\delta$ with $O(n^d)$ points, the number of edges would be bounded by $O(n^{d+1})$.

For each point $b \in B$ and each $i \in \{0, \delta, 2\delta\}$, the Voronoi cell $V_b^i$ has $O(n^{\lceil d/2 \rceil})$ vertices; since $3n$ such cells are used in the construction of the arrangement, the total number of Voronoi vertices in the arrangement is $O(n^{\lceil d/2 \rceil + 1})$. Next, we bound the number of intersection points of these Voronoi cells. For each $d$-tuple $(V_{b_1}^{i_1}, V_{b_2}^{i_2}, \ldots, V_{b_d}^{i_d})$ of Voronoi cells, for $d$ distinct points $b_1, \ldots, b_d$ and $d$ values $i_1, \ldots, i_d \in \{0, \delta, 2\delta\}$, let $v$ denote a point in the intersection of these cells. Similar to our analysis for 2-dimensional distributions, we show that $v$ lies on the weighted bisector of $b_1, \ldots, b_d$, where the weight of each point $b_j$ is $y(b_j) + i_j$. Since this weighted bisector is a straight line, and since a straight line intersects a convex polytope in at most 2 points, each $d$-tuple of Voronoi cells intersect in at most 2 points, where the number of such $d$-tuples are $O(n^d)^4$. Therefore, we conclude that the number of intersection points is $O(n^d)$, as desired.

Let $v$ be an intersection point of the $d$ Voronoi cells $V_{b_1}^{i_1}, V_{b_2}^{i_2}, \ldots,$ and $V_{b_d}^{i_d}$. Suppose $v$ lies on the $(d-1)$-dimensional hyperplane of $V_{b_1}^{i_1}$ that is shared between $b_1$ and a point $b_{d+1} \in B$. In this case,

$$\mathrm{d}_{y_{b_1}^{i_1}}(v, b_1) = \mathrm{d}_{y_{b_1}^{i_1}}(v, b_{d+1}) = \min_{b \in B} \mathrm{d}_{y_{b_1}^{i_1}}(v, b). \tag{7}$$

We claim that for any $k \in \{1, \ldots, d\}$, $y_{b_k}^{i_k}(b_{d+1}) = y(b_{d+1})$. Consider the following two cases:

- If $b_{d+1} = b_j$ for some $j \in \{2, \ldots, d\}$, then
$$\mathrm{d}_y(v, b_1) - i_1 = \mathrm{d}_{y_{b_1}^{i_1}}(v, b_1) = \mathrm{d}_{y_{b_1}^{i_1}}(v, b_j) = \mathrm{d}_y(v, b_j),$$

  and also
$$\mathrm{d}_y(v, b_j) - i_j = \mathrm{d}_{y_{b_j}^{i_j}}(v, b_j) = \mathrm{d}_{y_{b_j}^{i_j}}(v, b_1) = \mathrm{d}_y(v, b_1).$$

  Therefore, $i_1 = i_j = 0$ and for any $k \in \{1, \ldots, d\}$, $y_{b_k}^{i_k}(b_{d+1}) = y(b_{d+1})$.
- Otherwise, $b_{d+1} \notin \{b_1, \ldots, b_d\}$ and by the construction of $y_{b_k}^{i_k}(\cdot)$, for any $k \in \{1, \ldots, d\}$, $y_{b_k}^{i_k}(b_{d+1}) = y(b_{d+1})$.

Therefore, from Equation (7),

$$\mathrm{d}_y(v, b_{d+1}) = \mathrm{d}_{y_{b_1}^{i_1}}(v, b_{d+1}) = \min_{b \in B} \mathrm{d}_{y_{b_1}^{i_1}}(v, b) = \min_{b \in B} \mathrm{d}_y(v, b), \tag{8}$$

and for any $k \in \{1, \ldots, d\}$, since $v$ lies on a $(d-1)$-dimensional hyperplane of $V_{b_k}^{i_k}$, it has to lie on the hyperplane of $V_{b_k}^{i_k}$ that is shared between $b_k$ and $b_{d+1}$; therefore, $\mathrm{d}(v, b_{d+1}) - y(b_{d+1}) = \mathrm{d}(v, b_k) - (y(b_k) + i_k)$. Thus,

$$\mathrm{d}(v, b_1) - (y(b_1) + i_1) = \mathrm{d}(v, b_2) - (y(b_2) + i_2) = \ldots = \mathrm{d}(v, b_d) - (y(b_d) + i_d). \tag{9}$$

From Equation (9), the point $v$ lies on the weighted bisector of $b_1, \ldots, b_d$ with weights $y(b_1) + i_1, \ldots, y(b_d) + i_d$, and the $d$-tuple $(V_{b_1}^{i_1}, V_{b_2}^{i_2}, \ldots, V_{b_d}^{i_d})$ intersect each other in at most 2 points.

**Lemma A.1.** *For any dimension $d \geq 2$, a parameter $\delta > 0$, and a $\delta$-feasible transport plan $\hat{\tau}, y(\cdot)$, the residual graph $\mathcal{G}_\delta$ has $O(n^d)$ vertices and $O(n^{d+1})$ edges.*

**Compressing the transport plan.** Recall that for any semi-discrete transport plan $\tau$ from $B$ to $A$, a set of weights $y(\cdot)$, and a parameter $\delta > 0$, the compressed transport plan $\hat{\tau}$ is a discrete transport plan from $B$ to $A_\delta$. Note that the definition of $\delta$-feasibility (and more precisely, condition (F1)) is applicable for the discrete OT as well; the discrete transport plan $\hat{\tau}$ along with weights $y(\cdot)$ for $B$ is $\delta$-feasible if $\hat{\tau}$ transports the mass of each point $b \in B$ to points of $A_\delta$ within its $2\delta$-expanded Voronoi cell. Any complete transport plan $\hat{\tau}$ from $B$ to $A_\delta$ that is $\delta$-feasible along with weights $y(\cdot)$ is called $\delta$-optimal.

---

[4]Here, the constant is $O(n^d)$ hides a factor of $3^d$.

**Lemma A.2.** *Any $\delta$-optimal compressed transport plan $\hat{\tau}, y(\cdot)$ from $B$ to $A_\delta$ is $2\delta$-close.*

*Proof.* For any transport plan $\tau'$ from $B$ to $A_\delta$ define $\cancel{c}_y(\tau') := \sum_{(r,b)\in A_\delta \times B} \mathrm{d}_y(r,b)\tau'(r,b)$. Then,

$$\cancel{c}_y(\tau') = \sum_{(r,b)\in A_\delta \times B} \mathrm{d}_y(r,b)\tau'(r,b) = \sum_{(r,b)\in A_\delta \times B} \big(\mathrm{d}(r,b) - y(b)\big)\tau'(r,b) = \cancel{c}(\tau') - \sum_{b\in B} y(b)\nu(b).$$
(10)

For any point $r \in A_\delta$, let $b_r$ denote the weighted nearest neighbor of $r$, i.e., $b_r := \arg\min_{b\in B} \mathrm{d}_y(r,b)$. By the definition of $\delta$-feasibility, for any pair $(r,b)$ with $\hat{\tau}(r,b) > 0$, we have $\mathrm{d}_y(r,b) \le \mathrm{d}_y(r,b_r)+2\delta$. Therefore,

$$\cancel{c}_y(\hat{\tau}) = \sum_{(r,b)\in A_\delta \times B} \mathrm{d}_y(r,b)\hat{\tau}(r,b) \le \sum_{(r,b)\in A_\delta \times B} \big(\mathrm{d}_y(r,b_r) + 2\delta\big)\hat{\tau}(r,b)$$
$$= \sum_{(r,b)\in A_\delta \times B} \mathrm{d}_y(r,b_r)\hat{\tau}(a,b) + 2\delta.$$
(11)

Let $\tau^*$ denote an optimal transport plan from $B$ to $A_\delta$. Then,

$$\cancel{c}_y(\tau^*) = \sum_{(r,b)\in A_\delta \times B} \mathrm{d}_y(r,b)\tau^*(r,b) \ge \sum_{(r,b)\in A_\delta \times B} \mathrm{d}_y(r,b_r)\tau^*(r,b).$$
(12)

Combining Equations (11) and (12) and plugging $\hat{\tau}$ and $\tau^*$ in Equation (10),

$$\cancel{c}(\hat{\tau}) = \cancel{c}_y(\hat{\tau}) + \sum_{b\in B} y(b)\nu(b) \le \left[ \sum_{(r,b)\in A_\delta \times B} \mathrm{d}_y(r,b_r)\hat{\tau}(r,b) + 2\delta \right] + \sum_{b\in B} y(b)\nu(b)$$
$$\le \cancel{c}_y(\tau^*) + 2\delta + \sum_{b\in B} y(b)\nu(b) = \cancel{c}(\tau^*) + 2\delta.$$

Therefore, $\cancel{c}(\hat{\tau}) \le \cancel{c}(\tau^*) + 2\delta$ and $\hat{\tau}$ is $2\delta$-close. $\qquad\square$

The following observation, which is straightforward from the construction of the partitioning $\mathcal{X}_\delta$, helps in proving Lemmas 2.3 and 2.4.

**Observation A.3.** *For any region $\varphi \in \mathcal{X}_\delta$ and any point $b \in B$, the region $\varphi$ either completely lies inside $V_b^{2\delta}$ or it is completely outside $V_b^{2\delta}$.*

**Lemma 2.3.** *For any $\delta$-feasible semi-discrete transport plan $\tau, y(\cdot)$, the compressed transport plan $\hat{\tau}$ along with weights $y(\cdot)$ is also $\delta$-feasible.*

*Proof.* Consider any point $b \in B$ and any region $\varphi \in \mathcal{X}_\delta$ such that $\tau(\varphi, b) > 0$, i.e., in the compressed graph, $\hat{\tau}(r_\varphi, b) > 0$. To prove this lemma, we show that $r_\varphi \in V_b^{2\delta}$. Since $\tau(\varphi, b) > 0$, there exists a point $a \in \varphi$ such that $\tau(a,b) > 0$, and by the $\delta$-feasibility of $\tau, y(\cdot)$, we have $a \in V_b^{2\delta}$; therefore, using Observation A.3, the region $\varphi$ has to completely lie inside $V_b^{2\delta}$, and the representative point $r_\varphi$, which is a point inside $\varphi$ also lies inside $V_b^{2\delta}$. Hence, $\hat{\tau}, y(\cdot)$ is $\delta$-feasible. $\qquad\square$

**Lemma 2.4.** *Any $\delta$-feasible transport plan $\hat{\tau}, y(\cdot)$ from $B$ to $A_\delta$ can be converted into a $\delta$-feasible semi-discrete transport plan $\tau$ from $B$ to $A$.*

*Proof.* Consider any transport plan $\tau$ from $B$ to $A$ such that for any point $b \in B$ and any region $\varphi \in \mathcal{X}_\delta$, transports a mass of $\hat{\tau}(r_\varphi, b)$ from $b$ to the mass of $\mu$ inside the region $\varphi$. One such construction is to assign $\tau(a,b) = \frac{\hat{\tau}(r_\varphi, b)}{\mu(\varphi)}\mu(a)$ for each point $a \in \varphi$. We next show that the transport plan $\tau$ is $\delta$-feasible.

Consider any point $b \in B$ and any point $a \in A$ such that $\tau(a,b) > 0$. We prove this lemma by showing that $a \in V_b^{2\delta}$. Suppose $\varphi \in \mathcal{X}_\delta$ is the region containing the point $a$. Since $\tau(a,b) > 0$, we should have $\hat{\tau}(r_\varphi, b) > 0$, and by the $\delta$-feasibility of $\hat{\tau}, y(\cdot)$, we have $r_\varphi \in V_b^{2\delta}$. By Observation A.3, the whole region $\varphi$ has to lies inside $V_b^{2\delta}$ and therefore, the point $a$ also lies inside $V_b^{2\delta}$. Hence, $\tau, y(\cdot)$ is $\delta$-feasible. $\qquad\square$

**Augmentation.** In this section, we first show that upon augmenting a transport plan $\hat{\tau}$ along an augmenting path $P$, the transport plan remains valid. We then show that upon augmentation, either a backward edge gets removed from the transport plan or a free point will become fully transported. We finally prove Lemma 2.5.

For an augmenting path $P = \langle s, b_1, r_1, \ldots, r_k, t \rangle$ from a free point $b_1$ to a free point $r_k$, recall that the bottleneck capacity of $P$ is defined as

$$\mathrm{bc}(P) := \min\{\mathrm{ex}(b_1), \mathrm{ex}(r_k), \min_{i \in [1,k-1]} \hat{\tau}(r_i, b_{i+1})\},$$

where $\mathrm{ex}(b_1)$ (resp. $\mathrm{ex}(r_k)$) denotes the excess mass of $b_1$ (resp. $r_k$). To show that the transport plan after augmentation is a valid one, we show that no point $v$ in $\mathcal{G}_\delta$ transports more mass than the mass at $v$ and that each edge transports a non-negative amount of mass.

In the augmentation process, for any $i \in \{1, \ldots, k-1\}$, we decrease $\hat{\tau}(r_i, b_{i+1})$ by $\mathrm{bc}(P)$, where by definition, $\mathrm{bc}(P) \leq \hat{\tau}(r_i, b_{i+1})$; hence, $\hat{\tau}(r_i, b_{i+1}) \geq 0$ after augmentation. Furthermore, for any forward edge $(b_i, r_i)$, $i \in \{1, \ldots, k\}$, we increases $\hat{\tau}(r_i, b_i)$ by $\mathrm{bc}(P)$, and $\hat{\tau}(r_i, b_i)$ remains non-negative.

For any $i \in \{2, \ldots, k\}$, we increase $\hat{\tau}(b_i, r_i)$ (resp. decrease $\hat{\tau}(r_{i-1}, b_i)$) by $\mathrm{bc}(P)$; hence, the total amount of mass transported from $b_i$ remains unchanged. Similarly, for each $i \in \{1, \ldots, k-1\}$, we increase $\hat{\tau}(b_i, r_i)$ (resp. decrease $\hat{\tau}(r_i, b_{i+1})$) by $\mathrm{bc}(P)$ and the total amount of mass transported into $r_i$ remains unchanged. For the endpoint $b_1$ (resp. $r_k$), we only increase the amount of mass transported from $b_1$ (resp. into $r_k$) by $\mathrm{bc}(P)$, where by definition, $\mathrm{bc}(P) \leq \mathrm{ex}(b_1)$ (resp. $\mathrm{bc}(P) \leq \mathrm{ex}(r_k)$). Therefore, the total mass transport from $b_1$ (resp. into $r_k$) after augmentation is at most $\nu(b)$ (resp. $\mu_{r_k}$), as desired.

Note that if a backward edge $(r, b) \in P$ determines the bottleneck capacity of the augmenting path $P$, then $\hat{\tau}(r, b) = 0$ after augmentation and the backward edge is removed from the residual graph. Otherwise, if the endpoint $b_1$ (reps. $r_k$) determines the bottleneck capacity of $P$, then $b_1$ (resp. $r_k$) will be fully transported after augmentation.

**Lemma 2.5.** *The transport plan obtained after augmenting a $\delta$-feasible transport plan $\hat{\tau}, y(\cdot)$ along any augmenting path $P$ in the residual graph is $\delta$-feasible.*

*Proof.* Let $P = \langle s, b_1, r_1, \ldots, b_k, r_k, t \rangle$ denote an augmenting path in the residual graph. When augmenting the transport plan along $P$, we increase the mass transportation on forward edges $(b_i, r_i)$ for each $i \in [1, k]$ and decrease the mass transportation on the backward edges $(r_i, b_{i+1})$ for each $i \in [1, k-1]$. Therefore, any pair $(r, b)$ that transports mass after augmentation but was not transporting mass before augmentation has to be a forward edge of $P$. Since we only add forward edges from the point $b$ to the representative points in $V_b^{2\delta} \cap A_\delta$, the edge $(r, b)$ satisfies the $\delta$-feasibility condition (F1), and $\hat{\tau}, y(\cdot)$ remains feasible after augmentation. $\square$

# B    Missing Details of Section 3

In this section, we provide the missing details of the implementation of the INCREASEWEIGHTS and ACYCLIFY procedures and also prove the properties of the three procedures.

## B.1    Missing Proofs of the SEARCHANDAUGMENT Procedure

**Lemma 3.1.** *Suppose invariants (I1) and (I2) hold at the start of the* SEARCHANDAUGMENT *procedure. Then, during the execution of the* SEARCHANDAUGMENT *procedure,*

- *(S1) the transport plan $\hat{\tau}_\delta, y(\cdot)$ remains $\delta$-feasible,*
- *(S2) any point $b \in B$ (resp. backward edge $(r, b)$) marked as visited will not form an admissible augmenting path during the execution of the procedure, and*
- *(S3) there are no admissible cycles in the residual graph.*

*Proof.* We prove the properties separately in the following.

*Property (S1).* By the construction of the search path, any augmenting path computed by the SEARCHANDAUGMENT procedure is an admissible augmenting path. From Lemma 2.5, augmenting

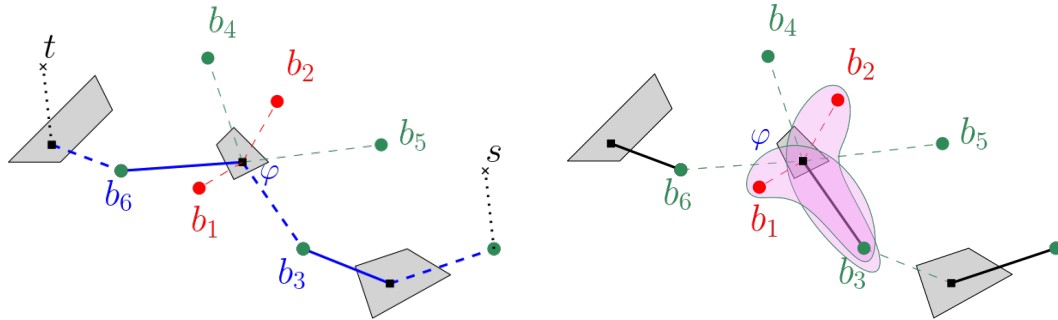

Figure 5: (left) An augmenting path found by the SEARCHANDAUGMENT procedure (blue path), and (right) two admissible triples (highlighted in pink) formed after augmentation.

$\hat{\tau}_\delta$ along an admissible augmenting path does not violate the $\delta$-feasibility condition; hence, the transport plan $\hat{\tau}_\delta, y(\cdot)$ is a $\delta$-feasible transport plan during the execution of the SEARCHANDAUGMENT procedure and (S1) holds.

Next, we present an overview of a new property of the SEARCHANDAUGMENT procedure, which we formally state and prove in Lemma B.1 below. Using that, we first prove (S3) and then prove (S2).

In Figure 5 (left), suppose the blue edges show an admissible augmenting path found by the SEARCHANDAUGMENT procedure, and suppose the green (resp. red) points show the unvisited (resp. visited) points of $B$. For the region $\varphi$, suppose $\langle b_1, b_2, \ldots, b_6 \rangle$ denote the set of neighbors of $r_\varphi$ in $\mathcal{G}_\delta$, sorted in increasing order of their weighted distance to $r_\varphi$. When the partial DFS procedure processes $r_\varphi$, the two weighted nearest neighbors of $\varphi$ (i.e., $b_1$ and $b_2$) are already marked as visited, leading the procedure to add $b_3$ to the search path. After augmentation (Figure 5 (right)), for the newly created backward edge $(r_\varphi, b_3)$, the only admissible triples containing $(r_\varphi, b_3)$ are $(b_1, r_\varphi, b_3)$ and $(b_2, r_\varphi, b_3)$ (the triples highlighted in pink), where $b_3$ is unvisited and both $b_1$ and $b_2$ are visited. More formally, as shown in Lemma B.1, for an augmenting path $P$ found by the procedure, assuming that (S3) holds before augmentation along $P$, for any newly created admissible triples $(b, r_\varphi, b')$ after augmenting the transport plan along $P$, the point $b$ (resp. $b'$) is marked as visited (resp. unvisited).

*Property (S3).* We begin by presenting an overview of our proof. Consider any augmenting path $P$ found by the SEARCHANDAUGMENT procedure. Assuming that (S3) holds before augmentation along $P$, all vertices that are reachable from a visited point $b$ by an admissible path in the procedure are also visited, since those points were also added to the search path, did not lead to an admissible augmenting path, marked as visited and removed from the search path. Hence, all points having an admissible path to the visited points (i.e., all points that are reachable from the visited points in our backward DFS) are also visited. Therefore, there are no admissible paths from an unvisited point to a visited point in the residual graph. After augmenting along $P$, by Lemma B.1, for any newly formed admissible triple $(b, r_\varphi, b')$, the point $b$ (resp. $b'$) is visited (resp. unvisited), and since there are no admissible paths from any unvisited point to any visited point, the newly formed admissible triple $(b, r_\varphi, b')$ does not form a cycle of admissible triples. Hence, (S3) holds after augmentation as well. We provide the details of the proof below.

Let $P^1, \ldots, P^k$ denote the sequence of augmenting paths computed by the SEARCHANDAUGMENT procedure, and let $\hat{\tau}_\delta^0, \hat{\tau}_\delta^1, \ldots, \hat{\tau}_\delta^k$ denote the sequence of transport plans computed by the procedure, i.e., $\hat{\tau}_\delta^0$ is the transport plan computed in the previous iteration, and for each $i \in [1, k]$, $\hat{\tau}_\delta^i$ is obtained by augmenting $\hat{\tau}_\delta^{i-1}$ along $P^i$. Let $\mathcal{G}^i$ denote the residual graph corresponding to $\hat{\tau}_\delta^i$ for each $i \in [0, k]$. Let $V^i$ (resp. $U^i$) denote the set of visited (resp. unvisited) points in $B$ when the procedure augments $\hat{\tau}_\delta^{i-1}$ along $P^i$.

Initially, from invariant (I2), there are no cycles of admissible triples in the residual graph, and (S3) holds for $\mathcal{G}^0$. For any $i \in [1, k]$, assuming $\mathcal{G}^0, \ldots, \mathcal{G}^{i-1}$ satisfies (S3), we show that $\mathcal{G}^i$ also satisfies (S3). Suppose $(b, r, b')$ is any admissible triple in $\mathcal{G}^i$ formed after augmenting $\hat{\tau}_\delta^{i-1}$ along $P^i$, i.e., the triple $(b, r, b')$ is admissible in $\mathcal{G}^i$ but not in $\mathcal{G}^{i-1}$. We show that the triple $(b, r, b')$ does not participate in any admissible cycles; hence, using property (S3) on $\mathcal{G}^{i-1}$, there are no admissible cycles in $\mathcal{G}^i$ and property (S3) holds for $\mathcal{G}^i$ as well.

Since (S3) holds for $\mathcal{G}^0, \ldots, \mathcal{G}^{i-1}$, for any visited point $b \in V^{i-1}$, the point $b$ has been added to the search path $Q$ by the SEARCHANDAUGMENT procedure, did not lead to the computation of an augmenting path, marked as visited and removed from the path. In this case, any point $b''$ that is reachable from $b$ by an admissible path in $\overleftarrow{\mathcal{G}^{i-1}}$ (and therefore, is reachable by our partial DFS procedure from $b$) would have also been added to the path, marked as visited and removed from $Q$, i.e., any point $b'' \in B$ that has an admissible path to the visited point $b$ in the residual graph $\mathcal{G}^{i-1}$ is also visited. By this observation, there are no admissible paths from any unvisited point in $B$ to any visited point in $\mathcal{G}^{i-1}$. From Lemma B.1, for any newly formed admissible triple $(b, r, b')$, we have $b \in V^{i-1}$ and $b' \in U^{i-1}$. Thus, all admissible triples formed after augmenting $\hat{\tau}_\delta^{i-1}$ along $P^i$ are from a visited point to an unvisited point, while there are no admissible paths from any unvisited point to any visited point; therefore, the newly formed admissible triples do not form any admissible cycles and (S3) holds for $\mathcal{G}^i$ as well.

*Property (S2).* We use the property (S3) to show that (S2) holds. For any point $b \in B$ that is marked as visited, as discussed above, if (S3) holds, all vertices that are reachable from $b$ by an admissible path in our backward DFS (i.e., all points having an admissible path to $b$ in the residual graph) are also visited. Since any free point $b_f \in B$ is unvisited, $b_f$ does not have an admissible path to any visited point $b \in B$ and therefore, the visited points do not participate in an admissible augmenting path. Furthermore, the procedure marks a backward edge $(r, b)$ as visited if, for each admissible triple $(b', r, b)$, the point $b'$ is visited. Since the point $b'$ cannot be included in an admissible augmenting path, the visited backward edge $(r, b)$ also does not form an admissible augmenting path.

$\square$

**Lemma B.1.** *During the execution of the* SEARCHANDAUGMENT *procedure, suppose $P$ is an admissible augmenting path found by the procedure, and let $\mathcal{G}$ (resp. $\mathcal{G}'$) denote the residual graph before (resp. after) augmenting the transport plan along $P$. Let $(b, r, b')$ denote an admissible triple in $\mathcal{G}'$ that is not admissible in $\mathcal{G}$. Assuming that (S3) holds prior to augmentation along $P$, the point $b$ is marked as visited, and $b'$ is marked as unvisited.*

*Proof.* Consider any triple $(b, r, b')$ that is admissible in $\mathcal{G}'$ but not in $\mathcal{G}$. Recall that by the definition of the admissible triples, $(r, b')$ is a backward edge in $\mathcal{G}'$ and $\mathrm{d}_y(r, b) > \mathrm{d}_y(r, b')$. Since the SEARCHANDAUGMENT procedure does not change the weights $y(\cdot)$, the only case where $(b, r, b')$ is not admissible in $\mathcal{G}$ is when $(r, b')$ is not a backward edge in $\mathcal{G}$, i.e., the pair $(b', r)$ is in $P$ as a forward edge, and augmenting $\hat{\tau}_\delta$ along $P$ results in transporting mass from $b'$ to $r$. On the other hand, by step 2(a) of the SEARCHANDAUGMENT procedure, a forward edge $(b', r)$ will be added to the search path only if $b'$ is the weighted nearest unvisited neighbor of $r$; in other words, since $\mathrm{d}_y(r, b) > \mathrm{d}_y(r, b')$ and the procedure added $b'$ to the search path (instead of $b$), the point $b$ was marked as visited by the procedure. Therefore, for any newly formed admissible triple $(b, r, b')$, point $b$ (resp. $b'$) is marked as visited (resp. unvisited). $\square$

### B.2 Missing Details and Proofs of the INCREASEWEIGHTS Procedure

After the execution of the SEARCHANDAUGMENT procedure, no admissible augmenting paths remain in the residual graph, i.e., there are no admissible paths from the source vertex $s$ to the sink vertex $t$. The INCREASEWEIGHTS procedure increases the weights of the subset of points in $B$ that are reachable from $s$ by admissible paths to expand their Voronoi cells and to create new admissible triples in the residual graph. For instance, in Figure 6 (left), the path from $s$ to $t$ is not admissible (the triple $(b_2, r_2, b_3)$ is not admissible, as $b_3$ has a lower weighted distance to $r_2$ than $b_2$). As shown in Figure 6 (right), the INCREASEWEIGHTS procedure then increases the weight of the points $b_1$ and $b_2$ (which are reachable from $s$), leading to the formation of an admissible augmenting path (note that upon updating the weights, the regions corresponding to $r_1$ and $r_2$ have slightly changed). The details of the INCREASEWEIGHTS procedure are described below.

For each point $r \in A_\delta$, let $\mathcal{N}(r) \subseteq B$ denote the set of points $b \in B$ with $\hat{\tau}_\delta(r, b) > 0$, sorted in decreasing order of their weighted distance to $r$, i.e., $\mathcal{N}(r) = \{b_1, \ldots, b_k\}$ where $\mathrm{d}_y(r, b_1) \geq \mathrm{d}_y(r, b_2) \geq \ldots \mathrm{d}_y(r, b_k)$. Mark all points $b \in B$ and all forward edges $(b, r)$ as unvisited and set $\mathcal{K} = \emptyset$, let $U = B$ denote the set of unvisited points of $B$, and define $Q := \langle s \rangle$ as the search path that the algorithm grows. Execute the following steps until $Q$ becomes empty.

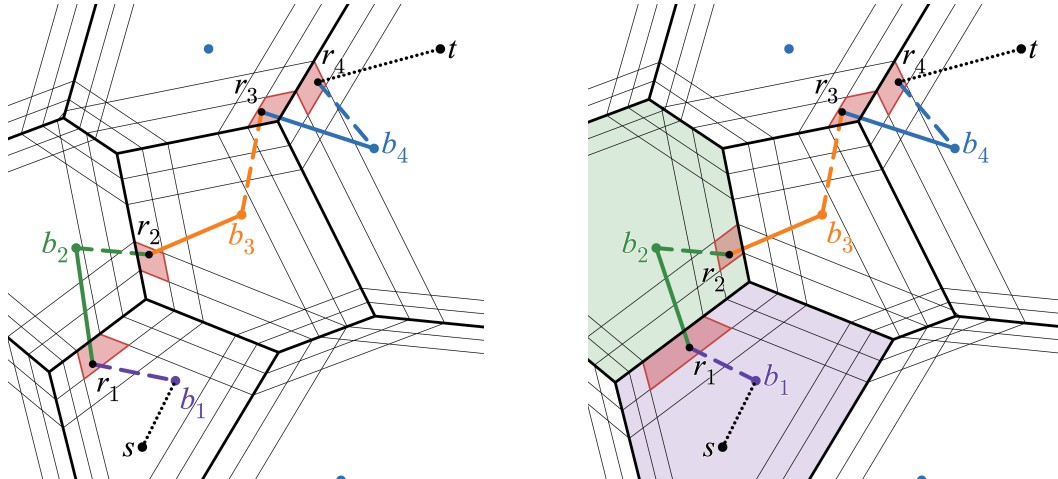

Figure 6: (left) After the execution of the SEARCHANDAUGMENT procedure, there are no admissible augmenting paths in $\mathcal{G}_\delta$, and (right) by increasing the weights of the points that are reachable from $s$ by augmenting paths (points $b_1$ and $b_2$), new admissible triples are created (e.g. $(b_2, r_2, b_3)$), which might lead to the formation of admissible augmenting paths.

1. If $Q = \langle s \rangle$, then if there exists an unvisited point $b \in U$ such that $(s, b) \in \mathcal{G}_\delta$, then add $b$ to $Q$ as $b_1$. Otherwise, remove $s$ from $Q$.

2. If $Q = \langle s, b_1, r_1, \ldots, b_i \rangle$ for some $i \geq 1$,

    (a) If there exists an unvisited forward edge $(b_i, r)$ in $\mathcal{G}_\delta$, add $r$ to $Q$ as $r_{i+1}$.

    (b) Otherwise, mark $b_i$ as visited, remove $b_i$ from $U$, add $b_i$ to $\mathcal{K}$, and remove $b_i$ from $Q$.

3. If $Q = \langle s, b_1, r_1, \ldots, b_i, r_i \rangle$ for some $i \geq 1$, let $b := \arg\min_{b' \in U \cap \mathcal{N}(r)} \mathrm{d}_y(r_i, b')$ denote the unvisited point of $b$ with the minimum weighted distance to $r$ among all points of $B$ that transport mass to $r$.

    (a) If $(b_i, r_i, b)$ is admissible, i.e., $\mathrm{d}_y(r_i, b) > \mathrm{d}_y(r_i, b_i)$, then add $b$ as $b_{i+1}$ to $Q$.

    (b) Otherwise, remove $r_i$ from $Q$ and mark $(b_i, r_i)$ as visited.

After the DFS procedure terminates, for each point $b \in \mathcal{K}$, set $y(b) \leftarrow y(b) + \delta$. This completes the description of the DFS step. We next describe how to recompute the residual graph and the compressed transport plan with respect to the updated weights.

Let $y(\cdot)$ (resp. $y'(\cdot)$) denote the weights of the points in $B$ after (resp. before) the weight updates, let $\mathcal{X}_\delta$ (resp. $\mathcal{X}'_\delta$) denote the partitioning of the set $A$ with respect to weights $y(\cdot)$ (resp. $y'(\cdot)$), and let $A_\delta$ (resp. $A'_\delta$) denote the set of representative points of the regions in $\mathcal{X}_\delta$ (resp. $\mathcal{X}'_\delta$). Furthermore, let $\hat{\tau}'_\delta$ denote the transport plan maintained by the algorithm for partitioning $\mathcal{X}'_\delta$. To compute the new transport plan $\hat{\tau}_\delta$ for the point set $A_\delta$, the INCREASEWEIGHTS procedure first computes the arrangement $\mathcal{Y}$ of all $3n$ cells used to construct $\mathcal{X}'_\delta$ with all $3n$ cells used to construct $\mathcal{X}_\delta$, i.e., $\mathcal{Y}$ is the arrangement of Voronoi cell, $\delta$-expanded Voronoi cell, and $2\delta$-expanded Voronoi cell of each point $b \in B$ both before and after weight updates. See Figure 7. For each region $\varphi \in \mathcal{X}'_\delta \cup \mathcal{X}_\delta$, let $\mathcal{C}(\varphi) \subseteq \mathcal{Y}$ denote the set of regions of $\mathcal{Y}$ that lie inside $\varphi$. For each region $\varrho \in \mathcal{Y}$, pick an arbitrary representative point $r_\varrho$ inside $\varrho$. We denote the set of all representative points of the regions in $\mathcal{Y}$ by $Y$.

The INCREASEWEIGHTS procedure first converts $\hat{\tau}'_\delta$ to a transport plan $\hat{\tau}$ over the finer partitioning $Y \times B$ by simply splitting each region $\varphi'$ in $\mathcal{X}'_\delta$ (and its mass transportation) to fine regions of $\mathcal{Y}$ inside $\varphi'$. The procedure then uses $\hat{\tau}$ to construct a transport plan $\hat{\tau}_\delta$ over $A_\delta \times B$ by merging the regions of $\mathcal{Y}$ (and their mass transportation) to regions of $\mathcal{X}_\delta$. The details are provided next.

For each point $b \in B$, each region $\varphi' \in \mathcal{X}'_\delta$ with $\hat{\tau}'_\delta(r_{\varphi'}, b) > 0$, and each $\varrho' \in \mathcal{C}(\varphi')$, set $\hat{\tau}(\varrho', b) = \frac{\mu(\varrho')}{\mu(\varphi')} \hat{\tau}'_\delta(r_{\varphi'}, b)$. This completes the description of the split step and the construction of $\hat{\tau}$. Next, the procedure constructs $\hat{\tau}_\delta$ by setting, for each point $b \in B$ and each region $\varphi \in \mathcal{X}_\delta$, $\hat{\tau}_\delta(r_\varphi, b) := \sum_{\varrho \in \mathcal{C}(\varphi)} \hat{\tau}(\varrho, b)$. The transport plan $\hat{\tau}_\delta$ is defined over $A_\delta \times B$.

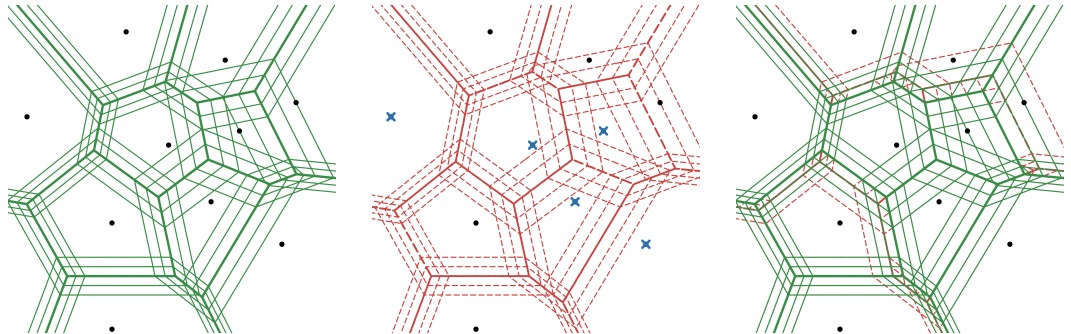

Figure 7: (left) The partitioning $\mathcal{X}'_\delta$ before updating weights, (middle) the partitioning $\mathcal{X}_\delta$ after updating the weights of the points of $B$ that are reachable from $s$ by admissible paths (the blue cross points), and (right) the combined partitioning $\mathcal{Y}$.

Finally, for each region $\varphi \in \mathcal{X}_\delta$, our algorithm stores a list $\mathcal{N}(r_\varphi)$ of all points $b \in B$ with $r_\varphi \in V_b^{2\delta}$, sorted in increasing order of their weights distance to $r_\varphi$, i.e., $\mathcal{N}(r_\varphi) = \langle b_1, \ldots, b_k \rangle, r_\varphi \in V_{b_i}^2 \delta$ for each $i \in [1, k]$, and $d_y(r_\varphi, b_i) \leq d_y(r_\varphi, b_j)$ for each $1 \leq i < j \leq k$. This completes the description of the INCREASEWEIGHTS procedure.

**Lemma 3.2.** *Suppose invariant (I1) holds at the start of the* INCREASEWEIGHTS *procedure. Then, during the execution of the* INCREASEWEIGHTS *procedure,*

- *(W1) the transport plan $\hat\tau_\delta, y(\cdot)$ remain $\delta$-feasible,*
- *(W2) the weight of each free point $b \in B$ increases by $\delta$, and*
- *(W3) the weight of each point $b \in B$ with free regions inside $V_b^\delta$ remains unchanged.*

*Proof.* Let $y(\cdot)$ (resp. $y'(\cdot)$) denote the weights of the points after (resp. before) the execution of the INCREASEWEIGHTS procedure, and let $\mathcal{X}_\delta$ (resp. $\mathcal{X}'_\delta$) be the partitioning with respect to weights $y(\cdot)$ (resp. $y'(\cdot)$). For any point $a \in A$, let $\varphi_a$ (resp. $\varphi'_a$) be the region in $\mathcal{X}_\delta$ (resp. $\mathcal{X}'_\delta$) that contains $a$, and let $b_a$ (resp. $b'_a$) denote the weighted nearest neighbor of $a$ in $B$ with respect to weights $y(\cdot)$ (resp. $y'(\cdot)$). For any pair of points $(a, b) \in A \times B$ with $\tau'_\delta(a, b) > 0$, by the $\delta$-feasibility of $\tau'_\delta, y'(\cdot)$, we have $d_{y'}(a, b) - 2\delta \leq d_{y'}(a, b'_a)$. To prove property (W1), we show that $d_y(a, b) - 2\delta \leq d_y(a, b_a)$.

Recall that the INCREASEWEIGHTS procedure finds the subset $\mathcal{K} \subset B$ of points that have admissible paths from the source vertex $s$ of the residual graph and increases the weights of all points in $\mathcal{K}$ by $\delta$. Consider the following cases:

- If $b \in \mathcal{K}$ is among the points whose weights are increased by the procedure, then

$$d_y(a, b) = d_{y'}(a, b) - \delta \leq d_{y'}(a, b'_a) + \delta \leq d_{y'}(a, b_a) + \delta \leq d_y(a, b_a) + 2\delta,$$

  where the second inequality holds from $\delta$-feasibility of $\tau'_\delta, y'(\cdot)$, the third inequality holds since $b_a$ is the weighted nearest neighbor of $a$ with respect to $y'(\cdot)$, and the last inequality holds since $y(b_a) \leq y'(b_a) + \delta$. Consequently, $d_y(a, b) - 2\delta \leq d_y(a, b_a)$.

- Otherwise, $b \notin \mathcal{K}$ and $d_y(a, b) = d_{y'}(a, b)$.

  - If $b_a \in \mathcal{K}$, then $d_{y'}(r_{\varphi_a}, b) \leq d_{y'}(r_{\varphi_a}, b_a)$ (since otherwise, the triple $(b_a, r_{\varphi_a}, b)$ would have been an admissible triple and $b_a \in \mathcal{K}$ would have resulted in $b \in \mathcal{K}$). In this case, since the weighted nearest neighbor of $a$ and $r_{\varphi_a}$ are the same,

$$d_{y'}(r_{\varphi_a}, b'_a) \leq d_{y'}(r_{\varphi_a}, b) \leq d_{y'}(r_{\varphi_a}, b_a) \leq d_{y'}(r_{\varphi_a}, b'_a) + \delta,$$

    where the last inequality holds since increasing the weight of $b'_a$ by $\delta$ made it the weighted nearest neighbor of $a$. Therefore, the region $\varphi_a$ and consequently, the point $a$ lie inside $V_b^\delta$ and $V_{b_a}^\delta$. Thus,

$$d_y(a, b) = d_{y'}(a, b) \leq d_{y'}(a, b_a) + \delta = d_y(a, b_a) + 2\delta.$$

  - Otherwise, if $b_a$ is also not in $\mathcal{K}$, then $d_y(a, b_a) = d_{y'}(a, b_a)$ and,

$$d_y(a, b) - 2\delta = d_{y'}(a, b) - 2\delta \leq d_{y'}(a, b_a) = d_y(a, b_a).$$

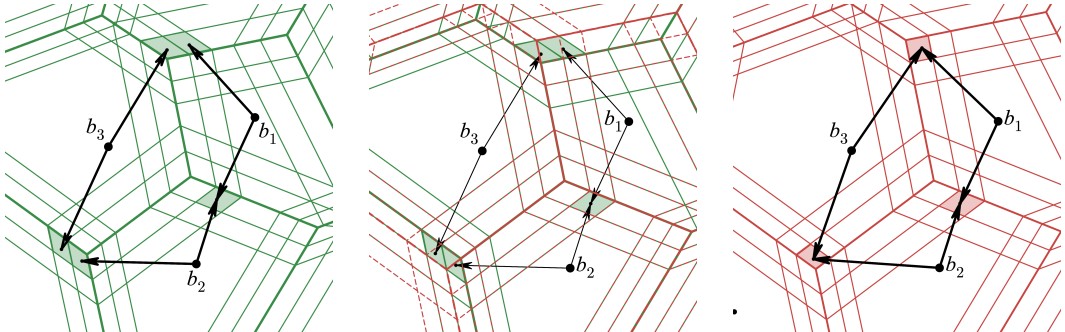

Figure 8: An example of a cycle in a transport plan that is created while increasing weights: (left) a transport plan that is a forest, (middle) the new Voronoi diagram and partitioning (red dashed lines) after increasing the weight of $b_1, b_2$, and $b_3$, and (right) a cycle formed in the new residual graph.

As a result, the transport plan $\tau_\delta$ along with the updated weights $y(\cdot)$ is $\delta$-feasible and property (W1) holds.

Note that each free point $b_f \in B$ has an edge from the source vertex $s$; therefore, $b_f \in \mathcal{K}$ and the INCREASEWEIGHTS increases the weight of $b_f$ by $\delta$, proving (W2). Furthermore, for each point $b \in B$ with a free region $\varphi \in V_b^\delta$, since we defined $\mathrm{d}_y(r_\varphi, t) = \min_{b' \in B} \mathrm{d}_y(r_\varphi, b')$, the triple $(b, r_\varphi, t)$ is admissible. Therefore, the SEARCHANDAUGMENT procedure should have added $r_\varphi$ and $b$ to the search path, not found an admissible augmenting path, and marked $b$ (resp. $(r, t)$) as visited. Therefore, there are no admissible paths from the source to $b$, or equivalently $b \notin \mathcal{K}$, and the weight of $b$ remains unchanged, leading to (W3). □

### B.3 Missing Details of the ACYCLIFY Procedure

The goal of the ACYCLIFY procedure is to ensure that the invariant (I2) holds, i.e., that at the beginning of each iteration of our algorithm, the transport plan is a forest and there are no admissible cycles in the residual graph. The procedure runs in three steps: (1) make $\hat{\tau}_\delta$ a forest, as described in Section B.3.1, (2) cancel any admissible cycles from the residual graph, as described in Section B.3.2, and (3) acyclify the transport plan again, as described in Section B.3.1. Note that our procedure acyclifies the transport plan (to make it a forest) twice, in steps (1) and (3). Making the transport plan a forest in the first step is essential for the efficiency of the second step, and making it a forest in the third step is essential for invariant (I2), as canceling admissible cycles might introduce cycles in the transport plan.

#### B.3.1 Acyclifying the Transport Plan

Similar to the Acyclify procedure introduced in [54, Section 3.3], we use a dynamic tree structure to make $\hat{\tau}_\delta$ a forest as follows. Let $\mathcal{E} = \langle e_1, e_2, \ldots, e_u \rangle$ denote the set of all edges $e = (r, b) \in A_\delta \times B$ with $\hat{\tau}_\delta(r, b) > 0$. For any $k \le u$, let $\mathcal{E}_k := \langle e_1, e_2, \ldots, e_k \rangle$. Define $F_0 := \emptyset$ as an empty forest and $\hat{\tau}'_0(r, b) = 0$ for all pairs $(r, b) \in A_\delta \times B$. Starting from $k = 1$, for any $k \le u$, the algorithm computes a forest $F_k$ and a transport plan $\hat{\tau}'_k$ defined over $F_k$ using $F_{k-1}$ and $\tilde{\tau}'_{k-1}$ as follows. If adding the edge $e_k$ to $F_{k-1}$ does not create a cycle, then the algorithm simply sets $F_k \leftarrow F_{k-1} \cup \{e_k\}$, $\hat{\tau}'_k(e_k) \leftarrow \hat{\tau}_\delta(e_k)$ and $\hat{\tau}'_k(e) \leftarrow \tau'_{k-1}(e)$ for all edges $e \in F_{k-1}$. Otherwise, adding $e_k$ to $F_{k-1}$ results in the creation of an even-length cycle $\mathcal{C}$. Let $c$ denote the minimum capacity of the edges in $\mathcal{C}$, and let $e^*$ denote the edge with the minimum capacity. Consider an ordering of the edges of the cycle $\mathcal{C}$ that starts with $e^*$, i.e., $\mathcal{C} = \langle e^* = e'_1, e'_2, \ldots, e'_{2j} \rangle$. The algorithm increases (resp. reduces) the mass transported along the edge $e'_{2i}$ (resp. $e'_{2i-1}$) by $c$ for each $i \in [1, j]$. Finally, the algorithm sets $F_k \leftarrow F_{k-1} \cup \{e_k\} \setminus \{e^*\}$. This completes the description of step 1. Using the dynamic tree structure by Sleator and Tarjan [56], each operation takes $O(\log n)$ amortized time, and since $|\mathcal{E}| = O(n^3)$, this process takes a total of $O(n^3 \log n)$ time.

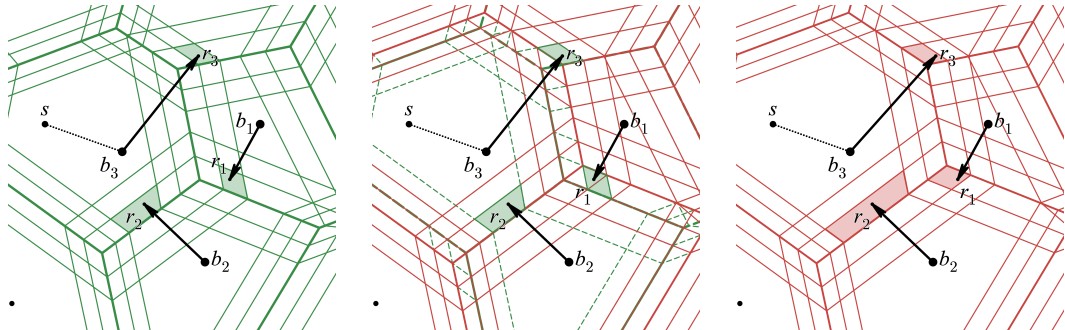

Figure 9: An example of the formation of admissible cycles due to updating the weights and the residual graph: (left) a transport plan with no admissible cycles in the corresponding residual graph, (middle) the new Voronoi diagram and partitioning (red lines) after increasing the weights of $b_3$ and $b_2$, and (right) an admissible cycle $\langle b_1, r_1, b_2, r_2, b_3, r_3 \rangle$ formed in the new residual graph.

### B.3.2 Acyclifying the Admissible Triples

To remove all admissible cycles from the residual graph, we use a partial DFS similar to the one described in the SEARCHANDAUGMENT procedure. Let $\overleftarrow{\mathcal{G}_\delta}$ be the graph formed by reversing the direction of all the edges of $\mathcal{G}_\delta$. The procedure first marks all points of $B$ and all backward edges as unvisited and defines $U := B$ as the set of unvisited points. While there exists an unvisited point $b \in B$, the procedure initializes a partial DFS by setting $Q = \langle b = b_1 \rangle$ and searches as follows until $Q$ becomes empty.

1. If $Q = \langle b_1, r_1, \ldots, b_i \rangle$ for some $i \geq 1$,

    (a) If there are no unvisited backward edges $(b_i, r)$ in $\overleftarrow{\mathcal{G}_\delta}$, then mark $b_i$ as visited and remove $b_i$ from $Q$ and $U$.
    (b) Otherwise, there exists an unvisited backward edge $(b_i, r_\varphi)$. Add $r_\varphi$ to $Q$ as $r_{i+1}$.

2. If $Q = \langle b_1, r_1, \ldots, b_i, r_i \rangle$ for some $i \geq 1$, let $b := \arg\min_{b' \in U} \mathrm{d}_y(r_i, b')$ be the unvisited point with the minimum weighted distance to $r_i$.

    (a) If $(b, r_i, b_i)$ is admissible, i.e., $\mathrm{d}_y(r_i, b) < \mathrm{d}_y(r_i, b_i)$,
        – If $b$ already exists in the path $Q$ as $b_j$, then $C = \langle b_j, r_j, \ldots, b_i, r_i, b_{i+1} = b = b_j \rangle$ is an admissible cycle. Cancel the cycle $C$ (as described below) and set $Q = \langle b_1, r_1, \ldots, b_j \rangle$.
        – Otherwise, add $b$ as $b_{i+1}$ to $Q$.
    (b) Otherwise, remove $r_i$ from $Q$ and mark the backward edge $(r_i, b_i)$ as visited.

Given an admissible cycle $C = \langle b_j, r_j, \ldots, b_i, r_i, b_{i+1} = b_j \rangle$, the procedure cancels the cycle as follows. Let $bc(C) := \min_{t \in [j,i]} \hat{\tau}_\delta(r_t, b_t)$ denote the bottleneck capacity of the cycle $C$. For any forward edge $(b_{t+1}, r_t)$ (resp. backward edge $(r_t, b_t)$) in $C$, set $\hat{\tau}_\delta(r_t, b_{t+1}) \leftarrow \hat{\tau}_\delta(r_t, b_{t+1}) + bc(C)$ (resp. $\hat{\tau}_\delta(r_t, b_t) \leftarrow \hat{\tau}_\delta(r_t, b_t) - bc(C)$). In this way, at least one of the backward edges of the cycle $C$ is removed from the residual graph and the cycle has vanished.

**Lemma 3.3.** *Suppose invariant (I1) holds at the start of the ACYCLIFY procedure. Then, during the execution of the ACYCLIFY procedure,*

(A1) *the transport plan $\hat{\tau}_\delta, y(\cdot)$ remains $\delta$-feasible, and*
(A2) *the transport plan $\hat{\tau}_\delta$ is a forest and there are no admissible cycles in the residual graph.*

*Proof.* In the first step of the ACYCLIFY procedure, the algorithm makes the transportation network to be a DAG. Note that the resulting transportation network is a subset of the transportation network before the ACYCLIFY procedure, and therefore, for any transporting edge $(r, b) \in A_\delta \times B$, the point $b$ is a $2\delta$-weighted nearest neighbor of $r$, i.e., the transport plan obtained after the first step of the ACYCLIFY procedure is $\delta$-feasible.

Next, we show property (A1) and (A2) for the second step of the procedure. By construction, all triples on the search path maintained by the procedure are admissible, and therefore, any cycle

computed by the second step of ACYCLIFY procedure is admissible. Since all forward edges on the computed cycles are from points $b \in B$ to regions in $V_b^{2\delta}$, canceling a cycle $C$ does not violate the $\delta$-feasibility condition (F1); hence, the transport plan $\hat{\tau}_\delta, y(\cdot)$ is a $\delta$-feasible transport plan during the execution of the second step of the ACYCLIFY procedure and (A1) holds.

To prove property (A2), we show that

(A3) any point $b \in B$ (resp. backward edge $(r, b)$) marked as visited will not form an admissible cycle during the execution of the procedure, and

(A4) during the execution of the second step of the ACYCLIFY procedure, the subgraph of the residual graph induced by visited vertices and their neighboring regions does not have any cycles of admissible triples. Furthermore, there are no admissible paths from an unvisited point to a visited point.

Assuming property (A3) holds, any point $b \in B$ that is marked as visited does not participate in an admissible cycle. Since the ACYCLIFY procedure stops when all points in $B$ are visited, there are no admissible cycles in the residual graph. Furthermore, since the transport plan is maintained by a dynamic tree structure the transport plan is a forest. Therefore, to prove property (A2), we show that (A3) holds.

Furthermore, the property (A3) is a direct corollary of property (A4), as explained next: For each visited point $b \in B$, all points $b' \in B$ having admissible paths to $b$ are also visited (note that the ACYCLIFY procedure searches on the residual graph in the reverse direction of the edges), and if (A4) holds, there are no admissible cycles solely formed by visited vertices; hence, $b$ does not form admissible cycles. Furthermore, the procedure marks a backward edge $(r, b)$ as visited if, for each admissible triple $(b', r, b)$, the point $b'$ is visited. Since $b'$ is not a part of any admissible cycles (assuming (A4) holds), the triple $(b', r, b)$ also cannot be a part of an admissible cycle. Therefore, to prove property (A3), we prove that (A4) holds in the following.

We use an inductive argument to prove (A4). Let $C^1, \ldots, C^k$ denote the sequence of admissible cycles found by the procedure, and let $\hat{\tau}_\delta^0, \ldots, \hat{\tau}_\delta^k$ denote the sequence of transport plans, where $\hat{\tau}_\delta^0$ is the transport plan maintained by the algorithm at the beginning of the second step of the procedure and $\hat{\tau}_\delta^i$ is obtained by canceling $\hat{\tau}_\delta^{i-1}$ along $C^i$. The property (A4) trivially holds at the beginning of the execution of the second step of the ACYCLIFY procedure.

Note that the ACYCLIFY procedure marks a point $b \in B$ as visited only if the search from $b$ did not lead to finding an admissible cycle, i.e., for all pairs $(b', r) \in B \times A_\delta$ such that $(b', r, b)$ is admissible, the point $b'$ is marked as visited and the backward edge $(r, b)$ is marked as visited. Hence, all points having an admissible path to the visited points (i.e., all points that are reachable from the visited points in our backward DFS) are also visited. Therefore, Assuming that property (A4) holds before marking $b$ as visited, property (A4) holds after marking $b$ as visited as well. Next, we show the same when we cancel a cycle $C^i$.

First, note that all points of $B$ that are on the search path are unvisited. Therefore, for any admissible cycle found by the procedure, the points of $B$ on the cycle are unvisited. Using an identical proof as in Lemma B.1, one can show that after canceling a cycle, for any newly formed admissible triple $(b, r, b')$, the point $b$ is visited and the point $b'$ is unvisited. In other words, canceling a cycle $C^i$ only creates additional admissible paths from visited points to unvisited points. Assuming that (A4) holds before canceling $C^i$, since no new admissible triples are created from a visited point to another visited point, the subgraph induced by visited points and their neighboring regions remain free of admissible cycles after canceling $C^i$. Furthermore, since no new admissible triples are created from an unvisited point to a visited point upon canceling $C^i$, there will be no admissible paths from an unvisited point to a visited point after cancellation; hence, (A4) remains satisfied. □

## C   Missing Details of Section 4

In this section, we analyze the efficiency of the three procedures SEARCHANDAUGMENT, INCREASEWEIGHTS, and ACYCLIFY.

## C.1 Efficiency of the SEARCHANDAUGMENT Procedure

The SEARCHANDAUGMENT procedure runs a partial DFS on the residual graph to find a set of admissible augmenting paths. The partial DFS procedure, upon backtracking from a point $b \in B$ (resp. $r \in A_\delta$), marks the point $b$ (resp. the backward edge $(b', r)$ used to reach $r$) as visited and does not visit the point $b$ (resp. the backward edge $(b', r)$) again in the same execution. Upon finding an augmenting path $P$, the procedure augments the transport plan along $P$ in $O(|P|)$ time. Let $\langle P_1, \ldots, P_k \rangle$ denote the set of all augmenting paths found by the SEARCHANDAUGMENT procedure. In 2 dimensions (resp. $d$ dimensions), since the residual graph has $O(n^3)$ edges (resp. $O(n^{d+1})$), the running time of the procedure would be $O(n^3 + \sum_{i=1}^k |P_i|)$ (resp. $O(n^{d+1} + \sum_{i=1}^k |P_i|)$). In Lemma C.1, we show that $\sum_{i=1}^k |P_i| = O(n^3)$ (resp. $\sum_{i=1}^k |P_i| = O(n^{d+1})$). Hence, each execution of the SEARCHANDAUGMENT procedure takes $O(n^3)$ (resp. $O(n^{d+1})$) time.

**Lemma C.1.** *The total length of augmenting paths computed during the execution of the* SEARCHAN-DAUGMENT *procedure is* $O(n^3)$ *in 2 dimensions and* $O(n^{d+1})$ *in d dimensions.*

*Proof.* Let $\hat{\tau}_\delta^0$ denote the transport plan maintained by the algorithm at the beginning of execution of the SEARCHANDAUGMENT procedure. To prove this lemma, we categorize the augmenting paths found by the procedure based on the source of their bottleneck capacity, namely (1) set $\mathcal{P}_v$ consisting of augmenting paths whose bottleneck capacity is determined based on the residual capacity of its endpoints, and (2) set $\mathcal{P}_e$ consisting of augmenting paths whose bottleneck capacity is determined based on mass transportation over its backward edges. We first show that $|\mathcal{P}_v| = O(n^2)$ and then show the same bound for $\mathcal{P}_e$. Since each augmenting path has a length of at most $2n$, we then conclude that the total length of all augmenting paths is $O(n^3)$.

Let $P$ be an augmenting path in $\mathcal{P}_v$. If the bottleneck capacity of $P$ is determined by a free point $b \in B$ (resp. free region $r \in A_\delta$), then the mass of $b$ (resp. $r$) will be fully transported after augmentation; therefore, since $|A_\delta \cup B| = O(n^2)$, we have $|\mathcal{P}_v| = O(n^2)$. Next, let $P$ be an augmenting path in $\mathcal{P}_e$; in this case, the backward edge $(r, b)$ determining the bottleneck capacity of $P$ will be removed from the transport plan after augmentation. Note that by Lemma B.1, for any newly formed admissible triples $(b', r, b)$, the point $b'$ is already marked as visited. By property (S2) in Lemma 3.1, the point $b'$ cannot form an admissible augmenting path during the same execution of the SEARCHANDAUGMENT procedure. Hence, the edge $(r, b)$ determining the bottleneck capacity of $P$ was a backward edge of the initial transport plan $\hat{\tau}_\delta^0$ and augmentation along each path $P \in \mathcal{P}_e$ removes one of the transporting edges of the transport plan $\hat{\tau}_\delta^0$. Using invariant (I2), $|\mathcal{P}_e| = O(n^2)$, as claimed. Hence, the total number of augmenting paths found by the procedure is $O(n^2)$, and since each augmenting path has a length of at most $2n$, their total length is $O(n^3)$.

Next, we extend our analysis to $d$ dimensional space, for any $d \geq 2$. Note that in $d$ dimensions, the residual graph has $O(n^d)$ vertices and $O(n^{d+1})$ edges. Hence, $|\mathcal{P}_v| = O(n^d)$. Since the transport plan $\hat{\tau}_\delta^0$ is a forest over the point set $A_\delta \cup B$, the total number of edges transporting a positive mass in $\hat{\tau}_\delta^0$ would be $O(n^d)$; since augmenting the transport plan along each augmenting path in $\mathcal{P}_e$ eliminates one of the edges transporting positive mass in $\hat{\tau}_\delta^0$, $|\mathcal{P}_e| = O(n^d)$. Finally, since each augmenting path has a length of at most $2n$, the total length of all augmenting paths found by the SEARCHANDAUGMENT procedure would be $O(n^{d+1})$. $\qquad\square$

## C.2 Efficiency of the INCREASEWEIGHTS Procedure

In this section, we show that for 2-dimensional (resp. $d$-dimensional) distributions, the IN-CREASEWEIGHTS procedure runs in $O(n^2(\Phi + n \log n))$ (resp. $O(n^d(\Phi + n \log n))$) time. The INCREASEWEIGHTS procedure runs a DFS that visits each edge of the residual graph at most once and has a total running time of $O(n^3)$ (resp. $O(n^{d+1})$). Furthermore, in the arrangement used to construct partitioning $\mathcal{Y}$, each point $b \in B$ has at most 6 Voronoi cells (three cells that are used in the construction of $\mathcal{X}_\delta$ and three that are used in the construction of $\mathcal{X}_\delta'$). Using a similar discussion as Section A, one can show that the total number of vertices in the arrangement used to construct $\mathcal{Y}$ is $O(n^2)$ (resp. $O(n^d)$), and the number of regions in $\mathcal{Y}$ is at most $O(n^2)$ (resp. $O(n^d)$). The construction of the transport plan $\hat{\tau}$ can be done in $O(n^2(\Phi + n))$ (resp. $O(n^d(\Phi + n))$) time since (1) the mass of all regions in $\mathcal{Y}$ can be determined in $O(n^2\Phi)$ (resp. $O(n^d\Phi)$) time (partitioning the regions in $\mathcal{Y}$ into simplices remains an arrangement with $O(n^2)$ (resp. $O(n^d)$) vertices and therefore,

have $O(n^2)$ (resp. $O(n^d)$) regions), and (2) the mass transported on each pair $(\varrho, b) \in \mathcal{Y} \times B$ can be determined in $O(1)$ time. Converting $\hat{\tau}$ to $\hat{\tau}_\delta$, as is done in the merge step, also takes $O(n^3)$ (resp. $O(n^{d+1})$) time, given that the total complexity of $\hat{\tau}$ is $O(n^3)$ (resp. $O(n^{d+1})$). Finally, storing a sorted list of neighbors for each region $r \in A_\delta$ takes $O(n^3 \log n)$ (resp. $O(n^{d+1}) \log n$) time in total. Hence, the execution of the INCREASEWEIGHTS procedure takes $O(n^2(\Phi + n \log n))$ (resp. $O(n^d(\Phi + n \log n))$) time.

**Lemma C.2.** *Each execution of the* INCREASEWEIGHTS *procedure takes $O(n^2(\Phi + n \log n))$ time in 2 dimensions and $O(n^d(\Phi + n \log n))$ time in $d$ dimensions.*

### C.3 Efficiency of the ACYCLIFY Procedure

The first step of this procedure uses a dynamic tree structure to acyclify the transport plan $\hat{\tau}_\delta$. Using the dynamic tree structure by Sleator and Tarjan [56], since the total number of edges of the graph is $O(n^3)$ in 2 dimensions (resp. $O(n^{d+1})$ in $d$ dimensions), the running time of this step would be $O(n^3 \log n)$ (resp. $O(n^{d+1} \log n)$). In the second step, the procedure runs a partial DFS procedure on the residual graph and cancels the admissible cycles. The partial DFS procedure, upon backtracking from a point $b \in B$ (resp. $r \in A_\delta$), marks the point $b$ (resp. the backward edge $(b', r)$ used to reach $r$) as visited and does not visit the point $b$ (resp. the edge $(b', r)$) again in the same execution. Furthermore, upon finding an admissible cycle $C$, it cancels the cycle in $O(|C|)$ time. Let $\langle C_1, \ldots, C_k \rangle$ denote the set of all cycles found in the execution of the ACYCLIFY procedure. In Lemma C.4, we show that $\sum_{i=1}^{k} |C_i| = O(n^3)$ in 2 dimensions (resp. $\sum_{i=1}^{k} |C_i| = O(n^{d+1})$ in $d$ dimensions). Given that the size of the residual graph is at most $O(n^3)$ (resp. $O(n^{d+1})$), the second step of the ACYCLIFY procedure takes a total of $O(n^3 + \sum_{i=1}^{k} |C_i|) = O(n^3)$ (resp. $O(n^{d+1})$) time, leading to the following lemma.

**Lemma C.3.** *Each execution of the* ACYCLIFY *procedure takes $O(n^3 \log n)$ time in 2 dimensions and $O(n^{d+1} \log n)$ time in $d$ dimensions.*

**Lemma C.4.** *The total length of admissible cycles computed during the execution of the second step of the* ACYCLIFY *procedure is $O(n^3)$ in 2 dimensions and $O(n^{d+1})$ time in $d$ dimensions.*

*Proof.* Let $\hat{\tau}_\delta^0$ denote the transport plan maintained by the algorithm at the beginning of execution of the ACYCLIFY procedure. To prove this lemma, we show that the ACYCLIFY procedure finds $O(n^2)$ admissible cycles, where each cycle has a length of at most $2n$; hence, the total length of all cycles found by the procedure would be $O(n^3)$.

Let $C$ be an admissible cycle found by the procedure; in this case, the backward edge determining the bottleneck capacity of $C$ will be removed from the transport plan after cancellation. For any admissible triple $(b, r, b')$ formed after canceling $C$, using an identical argument as in Lemma B.1, one can show that the edge $(r, b')$ is a backward edge that was on the cycle $C$ as a forward edge, and the point $b$ is marked as visited; hence, by Lemma 3.3, the point $b$ does not form an admissible cycle in the same execution of the ACYCLIFY procedure and therefore, the newly formed backward edge $(r, b')$ cannot be included in any admissible cycles. Therefore, each cycle cancellation removes one of the backward edges of $\hat{\tau}_\delta^0$, which is the transport plan obtained after the first step of the ACYCLIFY procedure, i.e., the transportation network of $\hat{\tau}_\delta^0$ is a forest and the number of its transporting edges is $O(n^2)$. Therefore, the total number of cycles found by the ACYCLIFY procedure is $O(n^2)$, and their total length is $O(n^3)$, as claimed.

We next show that the total length of admissible cycles in $d$ dimensions is $O(n^{d+1})$ in $d$ dimensions, for any $d \geq 2$. Note that in $d$ dimensions, the residual graph has $O(n^{d+1})$ edges. Since the transport plan $\hat{\tau}_\delta^0$ is a forest over the point set $A_\delta \cup B$, the total number of edges transporting a positive mass in $\hat{\tau}_\delta^0$ would be $O(n^d)$; since canceling each admissible cycle eliminates one of the edges transporting positive mass in $\hat{\tau}_\delta^0$, $|\mathcal{C}| = O(n^d)$. Since each admissible cycle has a length of at most $2n$, the total length of all admissible cycles found by the INCREASEWEIGHTS procedure would be $O(n^{d+1})$. $\square$

### C.4 Number of Iterations

**Lemma 4.1.** *For each scale $\delta$, the total number of iterations of step 2 of our algorithm is $O(n)$.*

*Proof.* Let $\tau_{2\delta}, y_{2\delta}(\cdot)$ denote the $2\delta$-feasible transport plan computed by our algorithm for scale $2\delta$, and let $\tau_\delta, y_\delta(\cdot)$ denote a partial transport plan maintained during the execution of step 2 of our algorithm. Let $\mathcal{X}_{2\delta}$ (resp. $\mathcal{X}_\delta$) denote the partitioning of the set $A$ with respect to weights $y_{2\delta}(\cdot)$ (resp. $y_\delta(\cdot)$). Let $\mathcal{Y}$ be the arrangement of all $3n$ cells used to construct $\mathcal{X}_{2\delta}$ with all $3n$ cells used to construct $\mathcal{X}_\delta$, i.e., $\mathcal{Y}$ is the arrangement of Voronoi cell, $\delta$-expanded Voronoi cell, and $2\delta$-expanded Voronoi cell of each point $b \in B$ with respect to weights $y_\delta(\cdot)$ along with the Voronoi cell, $2\delta$-expanded Voronoi cell, and $4\delta$-expanded Voronoi cell of each point $b \in B$ with respect to weights $y_{2\delta}(\cdot)$. For each region $\varrho \in \mathcal{Y}$, pick an arbitrary representative point $r_\varrho$ inside $\varrho$. We denote the set of all representative points of the regions in $\mathcal{Y}$ by $Y$. Let $\hat{\tau}_{2\delta}$ (resp. $\hat{\tau}_\delta$) denote the compressed transport plan for $\tau_{2\delta}$ (resp. $\tau_\delta$) using the partitioning $\mathcal{Y}$. Note that the partitioning $\mathcal{Y}$ is a refinement of both partitionings $\mathcal{X}_\delta$ and $\mathcal{X}_{2\delta}$. Define $\tau' := \hat{\tau}_{2\delta} - \hat{\tau}_\delta$.

We construct a bipartite graph $\mathcal{G}'$ over $Y \times B$, where for any pair $(r, b) \in Y \times B$, there exists an edge directed from $r$ to $b$ if $\tau'(r, b) < 0$ and directed from $b$ to $r$ if $\tau'(r, b) > 0$. Consider any directed path $P = \langle r_1, b_1, \ldots, r_k, b_k \rangle$ from a free region $r \in Y$ to a free point $b \in B$ (with respect to $\hat{\tau}_\delta$). The path $P$ is an augmenting path in the residual graph corresponding to $\hat{\tau}_\delta, y_\delta(\cdot)$.

Similar to the standard graph algorithms, we define the net-cost of the path $P$ as $\phi(P) := \sum_{i=1}^{k} d(r_k, b_k) - \sum_{i=1}^{k-1} d(r_k, b_{k+1})$. Let $b_0 := b_{r_1}$ be the weighted nearest neighbor of $r_1$. Then, we can rewrite the net-cost of $P$ as

$$\phi(P) = d(r_1, b_0) + \sum_{i=1}^{k} \left[ d(r_i, b_i) - d(r_i, b_{i-1}) \right]$$

$$= d_{y_\delta}(r_1, b_0) + \sum_{i=1}^{k} \left[ d_{y_\delta}(r_i, b_i) - d_{y_\delta}(r_i, b_{i-1}) \right] + y_\delta(b_k).$$

Due to $\delta$-feasibility of the transport plan $\hat{\tau}_\delta, y_\delta(\cdot)$, for all edges $(r_i, b_{i-1})$, $i \in [1, k]$, the point $b_{i-1}$ is a $2\delta$-weighted nearest neighbor of $r_i$; hence, $d_{y_\delta}(r_i, b_i) - d_{y_\delta}(r_i, b_{i-1}) \geq -2\delta$. Since the length of $P$ is at most $2n - 1$,

$$\phi(P) = d_{y_\delta}(r_1, b_0) + \sum_{i=1}^{k} \left[ d_{y_\delta}(r_i, b_i) - d_{y_\delta}(r_i, b_{i-1}) \right] + y_\delta(b_k) \geq d_{y_\delta}(r_1, b_0) + y_\delta(b_k) - 2n\delta. \quad (13)$$

Similarly, we can rewrite the net-cost of $P$ using weights $y_{2\delta}$ as follows.

$$\phi(P) = d_{y_{2\delta}}(r_1, b_0) + \sum_{i=1}^{k} \left[ d_{y_{2\delta}}(r_i, b_i) - d_{y_{2\delta}}(r_i, b_{i-1}) \right] + y_{2\delta}(b_k).$$

According to the $2\delta$-feasibility of $\hat{\tau}_{2\delta}, y_{2\delta}(\cdot)$, for all edges $(r_i, b_i)$, $i \in [1, k]$, the point $b_i$ is a $4\delta$-weighted nearest neighbor of $r_i$ (with respect to weights $y_{2\delta}(\cdot)$); hence, $d_{y_{2\delta}}(r_i, b_i) - d_{y_{2\delta}}(r_i, b_{i-1}) \leq 4\delta$. Since the length of $P$ is at most $2n - 1$,

$$\phi(P) = d_{y_{2\delta}}(r_1, b_0) + \sum_{i=1}^{k} \left[ d_{y_{2\delta}}(r_i, b_i) - d_{y_{2\delta}}(r_i, b_{i-1}) \right] + y_{2\delta}(b_k) \leq d_{y_{2\delta}}(r_1, b_0) + y_{2\delta}(b_k) + 4n\delta.$$

$$(14)$$

By property (W3) in Lemma 3.3, during the execution of step 2 of our algorithm, the weights of the points in $B$ with free regions in their $\delta$-expansion remain unchanged; therefore, since $b_0$ contains free regions inside its $\delta$-expanded Voronoi cell, $y_{2\delta}(b_0) = y_\delta(b_0)$ and $d_{y_{2\delta}}(r_1, b_0) = d_{y_\delta}(r_1, b_0)$. Combining with Equations (13) and (14),

$$y_\delta(b_k) - 2n\delta \leq \phi(P) - d_{y_\delta}(r_1, b_0) = \phi(P) - d_{y_{2\delta}}(r_1, b_0) \leq y_{2\delta}(b_k) + 4n\delta.$$

Equivalently,

$$y_\delta(b_k) - y_{2\delta}(b_k) \leq 6n\delta.$$

By property (W2) in Lemma 3.3, our algorithm increases the weight of the point $b_k$ by $\delta$ in each iteration while its mass is not fully transported by $\hat{\tau}_\delta$; therefore, the point $b_k$ cannot remain free after $6n$ iterations, i.e., after $O(n)$ iterations, there cannot be any remaining free points in $B$ and the step 2 of our algorithm terminates after $O(n)$ iterations. $\qquad \square$

