# OpenReview forum: "A Combinatorial Algorithm for the Semi-Discrete Optimal Transport Problem"
_NeurIPS.cc/2024/Conference — NeurIPS 2024 poster_

### Official Review · Reviewer_Eii6 · 2024-06-25

**Soundness:** 3
**Presentation:** 3
**Contribution:** 3
**Rating:** 8
**Confidence:** 2

**Summary:**

The paper presents a primal-dual algorithm for approximately solving the semi-discrete optimal transport problem. The algorithm runs $\log \Delta/\epsilon $ scales, starting with the scale $\delta = \Delta^2$ and halving it in each round. The idea is to maintain a $\delta$-feasible weight function during the course of the algorithm, which in each round constructs a residual graph from the three levels of expansion of the Voronoi diagram, and augments the transport plan along augmenting paths. To ensure the invariants and that the algorithm terminates, the algorithm considers only admissible paths and performs a procedure to eliminate cycles after each round. The proposed algorithm improves the running time from $n^9$ in Agarwal et al. to $n^4$ and the size of the graph from $n^5$ to $n^3$. The algorithm also extends to any dimension $d > 2$ and any $p\ge 1$-Wasserstein distance.

**Strengths:**

- The paper presents a set of strong results that significantly improve known results on semi-discrete optimal transport problem.
- Although I didn't verify all the proofs, I think the algorithm and procedures and the idea of using admissibility for DFS to avoid and remove cycles are all reasonable.

**Weaknesses:**

- There isn't any big weakness I can see.

**Questions:**

- Could the authors provide some more background on the construction of Voronoi diagram, ie, the complexity of the construction?
- Is there any lower bound on the runtime of algorithms for the semi-discrete OT problem?
- How is Theorem 1.2 compared with other algorithms for the discrete OT problem?

**Limitations:**

The authors adequately addressed the limitations of the paper.

---

> ### Author Rebuttal · Authors · 2024-08-06
>
> We appreciate your thorough review. We answer your questions below.
>
> ----
>
> >Could the authors provide some more background on the construction of Voronoi diagram, ie, the complexity of the construction?
>
> **Response:**
> For 2 dimensions, the weighted Voronoi diagram under the squared Euclidean distance (also known as the Laguerre diagram or the power diagram) can be constructed in $O(n\log n)$ time [1, 2]. For higher dimensions $d>2$, the construction time would be $O(n^{\lceil (d+1)/2\rceil})$ [3].
>
> ----
>
> [1] Fortune, Steven. "Voronoi diagrams and Delaunay triangulations." In Handbook of Discrete and Computational Geometry, pp. 705-721, 2017.
>
> [2] Sugihara, Kokichi. "Laguerre Voronoi diagram on the sphere." Journal for Geometry and Graphics 6, no. 1 (2002): 69-81.
>
> [3] Aurenhammer, Franz. "Power diagrams: properties, algorithms and applications." SIAM Journal on Computing 16, no. 1 (1987): 78-96.
>
> ----
> ----
>
> >Is there any lower bound on the runtime of algorithms for the semi-discrete OT problem?
>
> **Response:** In 2 dimensions, there are no known sub-quadratic algorithms even for the discrete version of the OT problem. The semi-discrete OT problem is a generalization of the discrete OT problem. So, to obtain a runtime better than $O(n^2)$, one would expect that a subquadratic-time algorithm for the discrete OT would come first.
>
> For higher dimensions, as stated in our paper, computing an $\varepsilon$-close semi-discrete transport plan in a time that is polynomial in $d$ and $\log 1/\varepsilon$ is known to be \#$P$-Hard [4].
>
> ----
> [4] Taşkesen, Bahar, Soroosh Shafieezadeh-Abadeh, and Daniel Kuhn. "Semi-discrete optimal transport: Hardness, regularization and numerical solution." Mathematical Programming 199, no. 1 (2023): 1033-1106.
>
> ----
> ----
>
> > How is Theorem 1.2 compared with other algorithms for the discrete OT problem?
>
> **Response:**
> Thank you for this question. Given $\mu$ (with support size $N$) and $\nu$ (with support size $k$), the execution times of the best-known discrete OT algorithm is almost-linear in the number of edges of the bipartite graph, i.e., $(Nk)^{1+o(1)}$ [5, 6].  In contrast, by preprocessing $\mu$ in near-linear time, we can compute a discrete OT plan for any query $\nu$ in $O(\sqrt{N}k^{4}\log 1/\delta)$ time. Thus, when $k$ is sufficiently small (say $k < N^{1/6}$), our algorithm is faster than all existing discrete OT algorithms. We will highlight this improvement to discrete OT algorithms in the next version of our paper.
>
> ----
>
> [5] Chen, Li, Rasmus Kyng, Yang P. Liu, Richard Peng, Maximilian Probst Gutenberg, and Sushant Sachdeva. "Maximum flow and minimum-cost flow in almost-linear time." FOCS 2022.
>
> [6] Agarwal, Pankaj K., Kyle Fox, Debmalya Panigrahi, Kasturi R. Varadarajan, and Allen Xiao. "Faster Algorithms for the Geometric Transportation Problem." SoCG 2017.

---

> > ### Comment · Reviewer_Eii6 · 2024-08-09
> >
> > I thank the authors for the response. I maintain my current score.

---

### Official Review · Reviewer_ZH4D · 2024-07-09

**Soundness:** 3
**Presentation:** 3
**Contribution:** 2
**Rating:** 6
**Confidence:** 3

**Summary:**

The paper studies the semi-discrete optimal transport problem, i.e. a generalization of bipartite matching where one side of the graph is not finite, but instead a probability distribution on some subset of the R^d.
It gives an algorithm that computes an optimal transport plan (up to some additive $\varepsilon) to such instances subject to an oracle that can integrate the density function of the continuous distribution in constant time over some given triangle.
The algorithm proposed is an adaption of the solution proposed by Agarwal et al. [SODA 24] which makes some improvements to depress the degree of the polynomial dependence of the runtime on the size of the discrete distribution in the input.
The authors further note that their methods yield a kind of sub-linear time online algorithm for discrete optimal transport where one (large) side of the bipartite graph is fixed and the smaller side is subject to updates. This is achieved by representing the larger side as a continuous distribution which can be efficiently sampled.

Note that I was not able to check the appendix where essentially all proofs live.

**Strengths:**

The paper is clear and easy to follow, and improves the state of the art on a very relevant topic (optimum transport).

**Weaknesses:**

- The work is largely a reworking of other recent results in the field, in particular the paper of Agarwal et al. at SODA24. It's not clear that improving the polynomial dependence here is highly relevant.

**Questions:**

No questions

**Limitations:**

Limitiations and impact have been adressed fully, this is a theoretical result with no negative consequences to be expected.

---

> ### Author Rebuttal · Authors · 2024-08-06
>
> We appreciate your thoughtful review. We address your concern below.
>
> > The work is largely a reworking of other recent results in the field, in particular the paper of Agarwal et al. at SODA24. It's not clear that improving the polynomial dependence here is highly relevant.
>
> **Response:** Improvement in the run time is only one part of our contribution. What we consider the most important contribution is the new algorithmic framework we propose for the semi-discrete OT problem.
> The algorithm from SODA 2024 discretizes a continuous distribution and then solves an instance of the discrete OT problem defined on these points in $\tilde{O}(n^9)$ time.
> In contrast, our algorithm extends the classical combinatorial primal-dual approach for discrete OT to the semi-discrete setting. It applies ideas such as augmenting paths on residual graphs directly to continuous regions (rather than samples from the regions). To our knowledge, our paper is the first to introduce this framework. As a result of the new combinatorial framework, we obtain a significant reduction in execution time to $\tilde{O}(n^4)$.
>
> We remark that polynomial improvements to discrete OT have had a significant impact on ML applications [1, 2, 3, 4].
> Similar improvements for the semi-discrete OT problem is an important challenge and we make significant progress in addressing this challenge.
>
>
> ----
> [1] Cuturi, Marco. "Sinkhorn distances: Lightspeed computation of optimal transport." NeurIPS 2013.
>
> [2] Altschuler, Jason, Jonathan Niles-Weed, and Philippe Rigollet. "Near-linear time approximation algorithms for optimal transport via Sinkhorn iteration." NeurIPS 2017.
>
> [3] Lahn, Nathaniel, Deepika Mulchandani, and Sharath Raghvendra. "A graph theoretic additive approximation of optimal transport." NeurIPS 2019.
>
> [4] Jambulapati, Arun, Aaron Sidford, and Kevin Tian. "A direct $\tilde{O}(1/\varepsilon)$ iteration parallel algorithm for optimal transport." NeurIPS 2019.

---

### Official Review · Reviewer_SH1u · 2024-07-13

**Soundness:** 3
**Presentation:** 4
**Contribution:** 4
**Rating:** 7
**Confidence:** 2

**Summary:**

This paper proposes a novel combinatorial algorithm for the problem known as semi-discrete optimal transport. The proposed method constructs a residual graph by considering the cells of a $\delta$-expanded Voronoi diagram, a relaxed concept of a weighted Voronoi diagram, as vertices. The algorithm performs augmentation on the residual graph while scaling $\delta$. This approach significantly reduces the theoretical computational complexity compared to existing methods. Furthermore, the proposed method can be applied to discrete optimal transport with large supports, enabling sublinear time responses to OT queries with respect to support size through preprocessing.

**Strengths:**

* The paper makes a significant contribution to the important problem of semi-discrete optimal transport in the field of machine learning.
* The proposed method drastically reduces computational complexity compared to existing methods. Additionally, its application to query responses through preprocessing also demonstrates excellent computational efficiency.
* The paper is exceptionally well-written, making it easy to follow the ideas despite the challenging content.

**Weaknesses:**

* Understanding that this is a theoretical paper, it is important to note that the lack of numerical experiments makes it difficult to assess the practical applicability of the proposed method.
* While this is not a weakness of the paper, I am not able to fully verify the validity or the details of the proofs and methodology, and therefore cannot guarantee their correctness.

**Questions:**

* Is this method an implementable and practically applicable algorithm, or is it challenging to apply in real-world scenarios due to large constant factors, thus possessing only theoretical value?

**Limitations:**

The authors properly address the limitations of the method within the main text.

---

> ### Author Rebuttal · Authors · 2024-08-06
>
> We appreciate your insightful review. We answer your question below.
>
> >Is this method an implementable and practically applicable algorithm, or is it challenging to apply in real-world scenarios due to large constant factors, thus possessing only theoretical value?
>
> **Response:**
> The main contribution of the paper is a new algorithmic framework for the semi-discrete OT problem, which we believe has much potential. Overall, our algorithm is practical and the constants hiding in the big-O notation are small. However, faithful implementation of our algorithm requires (a) constructing and dynamically maintaining arrangements of Voronoi diagrams, and (b) computing the exact mass inside any given triangle. Developing efficient and robust software that combines our algorithm with black-boxes (a) and (b) is a significant task. An interesting direction of future research is to explore which existing geometric software libraries, including the ones based on GPUs, can be used for our setting.
>
> Recollect that our algorithm does not make any assumptions on the smoothness of the continuous distribution.
> In future work, we will investigate whether we can significantly simplify our algorithm if we are willing to make certain smoothness assumptions (similar to the ones made in existing work [1, 2]) about the continuous distribution.
>
> ----
> [1] Oliker, Vladimir I and Laird D Prussner. ``On the numerical solution of the equation $\frac{\partial^{2}z}{\partial x^2} \frac{\partial^2 z}{\partial y^2} - \left (\frac{\partial^2 z}{\partial x \partial y} \right) = f$ and its discretizations, $i$.'' Numerische Mathematik, 54(3):271–293, 1989.
>
> [2] Merigot, Quentin, and Boris Thibert. "Optimal transport: discretization and algorithms." In Handbook of numerical analysis, vol. 22, pp. 133-212. Elsevier, 2021.

---

> > ### Comment · Reviewer_SH1u · 2024-08-09
> > **Comments on the rebuttal**
> >
> > I have reviewed the authors' response. My concerns have been resolved. I will maintain my current score.

---

### Decision · Program_Chairs · 2024-09-25

**Decision:**

Accept (poster)

**Comment:**

This paper presents a combinatorial algorithm that solves a well-studied task, known as semi-discrete optimal transport, where the goal is to compute the cheapest way to transport the mass from a continuous distribution to a discrete one (where the transport cost scales with the distance). This work, while building on recent algorithmic results for this problem, develops novel conceptual ideas to obtain a significantly faster algorithm. The reviewers agreed that this is an interesting contribution that is appropriate for presentation at NeurIPS.